# Conformal dispersion relations for defects and boundaries

Lorenzo Bianchi[a,b] [1], Davide Bonomi[c] [2]

[a] *Dipartimento di Fisica, Università di Torino and INFN - Sezione di Torino*
*Via P. Giuria 1, 10125 Torino, Italy*
[b] *I.N.F.N. - sezione di Torino,*
*Via P. Giuria 1, I-10125 Torino, Italy*
[c] *Department of Mathematics, City, University of London,*
*Northampton Square, EC1V 0HB London, United Kingdom*

## Abstract

We derive a dispersion relation for two-point correlation functions in defect conformal field theories. The correlator is expressed as an integral over a (single) discontinuity that is controlled by the bulk channel operator product expansion (OPE). This very simple relation is particularly useful in perturbative settings where the discontinuity is determined by a subset of bulk operators. In particular, we apply it to holographic correlators of two chiral primary operators in $\mathcal{N} = 4$ Super Yang-Mills theory in the presence of a supersymmetric Wilson line. With a very simple computation, we are able to reproduce and extend existing results. We also propose a second relation, which reconstructs the correlator from a double discontinuity, and is controlled by the defect channel OPE. Finally, for the case of codimension-one defects (boundaries and interfaces) we derive a dispersion relation which receives contributions from both OPE channels and we apply it to the boundary correlator in the O(N) critical model. We reproduce the order $\epsilon^2$ result in the $\epsilon$-expansion using as input a finite number of boundary CFT data.

---

[1] `lorenzo.bianchi@unito.it`
[2] `davide.bonomi@city.ac.uk`

# 1 Introduction and discussion

Extended excitations are important probes of Quantum Field Theories (QFT). On the one hand, the fact that a QFT must support extended excitations (such as Wilson lines for gauge theories) may provide additional constraints for the classification of consistent QFTs. On the other hand, it is interesting to understand how to carve out the space of consistent defects supported by a given QFT. Some of these questions can be made very precise in the context of defect Conformal Field Theories (dCFT) [1], where this rich interplay between bulk and defect finds an explicit realization in the defect crossing equation. The latter is the central ingredient of the defect bootstrap program, an ambitious endeavour which has recently expanded in various interesting directions, such as the study of line and surface defects in holographic theories [2–9], the classification of boundaries and defects in free theories [10–16], the analysis of general boundaries in CFTs and the application to statistical systems [17–27], the bootstrability program aimed to an exact solution of a defect CFT [28, 29] or the study of superconformal defects [30–45].

Many of these achievements were possible thanks to a large effort in developing the analytic techniques that have proven successful for bootstrapping standalone CFTs (see [46] for a review). In that context, the derivation of a Lorentzian inversion formula, allowing to extract the CFT data of a given correlator from its double discontinuity [47], has led to impressive results for CFTs which naturally admit an expansion in a small parameter, such as $1/N$ for holographic theories or $\epsilon$ for statistical models at the Wilson-Fisher fixed point. For a bulk two-point function in the presence of a defect there are two different OPE channels. The defect channel is controlled by the bulk-to-defect couplings and the scaling dimensions of the defect operators, while the bulk channel is controlled by the bulk OPE coefficients and the one-point function of the exchanged bulk operators. Correspondingly, *two Lorentzian inversion formulae* have been derived. One of them takes a single discontinuity that is controlled by the bulk OPE to extract the defect data [48], while the other takes a double discontinuity that is controlled by the defect OPE to extract bulk data [49].

One of the immediate consequences of these formulae is that all the information that is needed to reconstruct the correlator is encoded in its discontinuity (or double discontinuity). An important question is then how to reconstruct the correlator starting from its discontinuity. This is the content of a *dispersion relation*. The importance of dispersion relations in physics has been appreciated for a long time. Starting from the optical theorem, dispersion relations have then been derived for relativistic S-matrices and more recently for four-point correlation functions in CFTs [50]. These relations are particularly useful when the (double) discontinuity is simpler to compute than the whole correlator. Furthermore, they often enjoy important physical properties, such as positivity, which was used to derive an infinite set of dispersive sum rules [51].

In this paper we tackle the question of deriving a dispersion relation for dCFTs. The existence of two Lorentzian inversion formulae suggests that we should be able to write down two distinct dispersion relations, one controlled by the defect OPE and the other controlled

by the bulk OPE. It turns out that the latter is very simple to obtain, as the problem is effectively one-dimensional and it involves a single discontinuity [3]. Therefore, we will derive it using complex analysis together with the symmetries and the analytic structure of the correlator and we will confirm it by resumming the result of the Lorentzian inversion formula. Despite its simplicity (and thanks to it), this formula is still very useful as it allows to easily reproduce the full correlator in those cases where the discontinuity takes a simple form. In particular, since the discontinuity is dominated by the bulk OPE, this formula provides an explicit way to reconstruct the full defect correlator starting only from a subset of bulk data (the information about the defect is encoded in the one-point functions of bulk operators). Therefore, the formula is particularly suitable for those theories where the bulk is under great control and we can exploit that control to get information about the defect.

In this respect, a perfect example is the supersymmetric Wilson line in $\mathcal{N} = 4$ SYM, which will be our main application in this work. Perturbative results for the bulk two-point function at strong coupling were recently derived in [44] by using the Lorentzian inversion formula to extract the defect CFT data and then resumming the block expansion. We will reproduce this result with a single line integration, skipping the technically challenging intermediate steps.

One limitation of the Lorentzian inversion formula of [48] is that it fails to reproduce the CFT data of defect operators with low transverse spin. The meaning of low is related to a particular double lightcone limit of the correlator and, unlike the homogeneous case of [50], no bound is known on this behaviour, although in all known examples this value is lower than two. A similar limitation applies to the defect dispersion relation. In that case, however, it is easy to overcome this difficulty by introducing a suitable prefactor which improves the behaviour of the correlator in the relevant limit without altering its analytical properties.

The second dispersion relation should involve the double discontinuity and should be controlled by the defect channel. The derivation in this case is more involved, but for some specific values of the defect dimension one can relate the problem to the case without a defect [50] and, since the dispersion relation is a mathematical statement that is valid for any function of two complex variables with a given analytical structure, we propose a general formula. The formula is essentially analogous to the one derived in [51] and, in a similar way, it is technically hard to use. We test it only on disconnected correlators and we leave further checks of this formula for future work.

A slightly different discussion is needed for the case of codimension-one defects. In that case, the bulk two-point function depends on a single cross-ratio and the dispersion relation looks different. In particular, it is not possible to find a relation that is only controlled either by the bulk or by the boundary OPE. This is not so surprising as a similar drawback is present for the Lorentzian inversion formula derived in [22]. Therefore, we write down a dispersion relation which receives contributions from two different cuts, dominated by the two OPE

---

[3]A dispersion relation in terms of the single discontinuity has been used also for homogeneous CFTs [52], although in that case it cannot be derived from a Lorentzian inversion formula, which contains a double discontinuity.

channels. We show the effectiveness of this relation by applying it to the conformal boundary of the $O(N)$ critical model. In particular, we show that the results of [20] for the second-order $\epsilon$-expansion of the two-point correlator can be reproduced by the dispersion relation with a *finite number* of defect CFT data.

Our boundary formula involves two single discontinuities, while the Lorentzian inversion formula of [22] involves a single discontinuity controlled by the bulk channel and a double discontinuity controlled by the boundary. It would be interesting to understand whether a dispersion relation could be derived which exhibits the same features. This would be important because the double discontinuity is often more constraining than the single one. However the problem seems technically challenging as no closed form is known for the kernel of the Lorentzian inversion formula in [22].

**Note added:** While this paper was in preparation, we became aware of [53], whose content partially overlaps with the present work. We coordinated with the authors for a simultaneous submission.

## 2    Defect correlators and Lorentzian inversion formula

We consider a planar conformal defect of dimension $p$ and codimension $q$. We split the coordinates $x^\mu$ in $p$ parallel coordinates $x_\parallel^a$ and $q$ orthogonal coordinates $x_\perp^i$. The observable of interest for this work is the bulk two-point function of two identical bulk primary operators $\phi(x)$

$$\langle \phi(x_1)\phi(x_2)\rangle = \frac{F(z,\bar{z})}{|x_{1\perp}|^{\Delta_\phi}|x_{2\perp}|^{\Delta_\phi}} \tag{2.1}$$

which is determined up to a function of two conformal cross-ratios $z$ and $\bar{z}$. In Appendix A we summarize our conventions and the relation with different kinematical variables. As usual, we can understand the variables $z$ and $\bar{z}$ thinking of a defect in Euclidean kinematics stretched along $x_\perp = 0$. After using the preserved subgroup of conformal transformations to set $x_{1\parallel} = 0$ and $x_{1\perp} = (1,0\ldots0)$, the residual transformations can be used to fix the point $x_2$ on a plane with complex coordinates $z$ and $\bar{z}$. After Wick rotation, the complex coordinates $z$ and $\bar{z}$ are mapped into two real independent lightcone coordinates. The lightcones are then located at $z = 0,1$ and $\bar{z} = 0,1$ as shown in Figure 1

Any bulk two-point function admits two different OPE channels. In the bulk channel, controlled by the limit $z, \bar{z} \to 1$, the two operators are expanded in terms of an infinite tower of bulk operators through the ordinary bulk OPE. In the defect channel, for $z, \bar{z} \to 0$, both bulk operators are expanded in terms of defect operators using their defect OPE. These two channels are associated to two different block expansions.

The defect channel expansion reads

$$F(z,\bar{z}) = \sum_{\hat{\Delta},s} b_{\hat{\Delta},s}^2\, \hat{f}_{\hat{\Delta},s}(z,\bar{z}) \tag{2.2}$$

where the sum runs over defect primaries $\hat{\mathcal{O}}$ of dimension $\hat{\Delta}$ and transverse spin $s$ [4], and

---

[4]Defect primaries do not carry $SO(p)$ spin when the external operators are scalars [1].

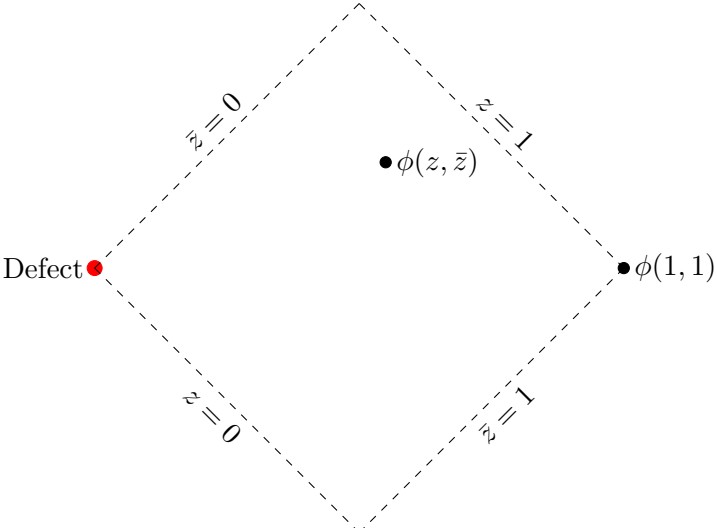

Figure 1: A Lorentzian plane orthogonal to the defect, which lies at the origin. The first operator lies at $z = \bar{z} = 1$, while the position of the second operator is parametrized by the real lightcone coordinates $z$ and $\bar{z}$.

$b_{\hat{\Delta},s}$ are the bulk-to-defect couplings associated to the two-point functions $\langle \phi \hat{\mathcal{O}} \rangle$. The defect conformal blocks in (2.2), eigenfunctions of the quadratic Casimir operator of the $SO(p+1,1) \times SO(q)$ group preserved by the defect, factorize accordingly and their exact form is [1]

$$\hat{f}_{\hat{\Delta},s}(r,w) = \hat{f}_{\hat{\Delta}}(r)\hat{g}_s(w) \tag{2.3}$$

with

$$\hat{f}_{\hat{\Delta}}(r) = r^{\hat{\Delta}} {}_2F_1\left(\hat{\Delta}, \frac{p}{2}, \hat{\Delta} + 1 - \frac{p}{2}, r^2\right), \quad \hat{g}_s(w) = w^{-s} {}_2F_1\left(-s, \frac{q}{2} - 1, 2 - \frac{q}{2} - s, w^2\right) \tag{2.4}$$

In (2.3), we introduced the convenient variables $r$ and $w$ defined by

$$z = rw \qquad\qquad \bar{z} = \frac{r}{w} \tag{2.5}$$

In the Euclidean regime, when $\bar{z} = z^*$, $r$ is the norm of the complex number $z$ and $w$ is the phase. In the Lorentzian case instead, when $z$ and $\bar{z}$ are independent real variables, also $r$ and $w$ are real. The $z \leftrightarrow \bar{z}$ symmetry is translated into

$$F(r,w) = F(r,1/w) \tag{2.6}$$

It is useful to notice that the angular part of the defect blocks, for integer $s$, is actually a Gegenbauer polynomial in the variable $\eta = \frac{1}{2}\left(w + \frac{1}{w}\right)$

$$\hat{g}_s(w) = \binom{s + \frac{q}{2} - 2}{\frac{q}{2} - 2}^{-1} C_s^{q/2-1}(\eta) \tag{2.7}$$

The bulk expansion reads

$$F(z,\bar{z}) = \left(\frac{\sqrt{z\bar{z}}}{(1-z)(1-\bar{z})}\right)^{\Delta_\phi} \sum_{\Delta,\ell} a_{\mathcal{O}} \lambda_{\phi\phi\mathcal{O}} f_{\Delta,\ell}(z,\bar{z}), \tag{2.8}$$

where $\Delta$ and $\ell$ are dimensions and spins of the operators $\mathcal{O}$ exchanged in the bulk OPE, $a_{\mathcal{O}}$ is the coefficient of the one-point function $\langle\mathcal{O}\rangle$ and $\lambda_{\phi\phi\mathcal{O}}$ is determined by the bulk three-point function $\langle\phi\phi\mathcal{O}\rangle$. The bulk conformal blocks $f_{\Delta,\ell}(z,\bar{z})$ are fully fixed by conformal invariance, although, generically, they do not admit a close form [54]. We spell out their expressions in Appendix A.

## 2.1 Lorentzian inversion formulae for defect CFT

The idea of the Lorentzian inversion formula for conformal field theories [47] is to provide a concrete and universal way to extract the defect CFT data from the double discontinuity of a four-point function (as opposed to the Euclidean inversion formula, which requires the knowledge of the full correlator and not only the double discontinuity). The powerful feature of that formula is that it allows for a cross talk between the different OPE channels, and very few operators in the $t$- and $u$-channel allow to extract an infinite tower of CFT data in the $s$-channel (the prototypical example being the identity operators which reconstruct the whole tower of double-twist operators in the crossed channel).

For the defect case, there are two possibilities. The first inversion formula was derived in [48] and it allows to extract the defect CFT data from a single discontinuity that is controlled by the bulk OPE. In particular the function $F(z,\bar{z})$, expressed in terms of the variables (2.5), can be written as

$$F(r,w) = \sum_{s=0}^{\infty}\int_{p/2-i\infty}^{p/2+i\infty}\frac{d\hat{\Delta}}{2\pi i}b(\hat{\Delta},s)\hat{g}_s(w)\hat{\Psi}_{\hat{\Delta}}(r) \tag{2.9}$$

with

$$\hat{\Psi}_{\hat{\Delta}}(r) = \frac{1}{2}\left(\hat{f}_{\hat{\Delta}}(r) + \frac{\hat{K}_{p-\hat{\Delta}}}{\hat{K}_{\hat{\Delta}}}\hat{f}_{p-\hat{\Delta}}(r)\right) \qquad\qquad \hat{K}_{\hat{\Delta}} \equiv \frac{\Gamma(\hat{\Delta})}{\Gamma(\hat{\Delta}-\frac{p}{2})} \tag{2.10}$$

The coefficient function $b(\hat{\Delta},s)$ encodes in a simple way the CFT data of the exchanged defect operators. Indeed, $b(\hat{\Delta},s)$ has simple poles corresponding to the spectrum of exchanged operators with residues given by the defect OPE coefficients $b_{\hat{\Delta},s}^2$ [48]. One can easily invert equation (2.9) using the orthogonality properties of $\hat{g}_s(w)$ and $\hat{\Psi}_{\hat{\Delta}}(r)$ thus finding a Euclidean inversion formula. However, it turns out that one does not need the knowledge of the full correlator $F(r,w)$ to compute $b(\hat{\Delta},s)$, but only of its discontinuity through the cut running from $w=0$ to $w=r$

$$\mathrm{Disc}F(r,w) = F(r,w+i0) - F(r,w-i0) \tag{2.11}$$

This is the content of the Lorentzian inversion formula

$$b(\hat{\Delta},s) = -\frac{\hat{K}_{\hat{\Delta}}}{i\pi\hat{K}_{p-\hat{\Delta}}}\int_0^1 dr\int_0^r dw\,\hat{\mu}(r,w)\,\hat{g}_{2-q-s}(w)\hat{\Psi}_{\hat{\Delta}}(r)\mathrm{Disc}F(r,w) \tag{2.12}$$

where the integration measure is

$$\hat{\mu}(r,w) = w^{1-q}(1-w^2)^{q-2}r^{-p-1}(1-r^2)^p \tag{2.13}$$

This formula was derived in [48] through a contour deformation of the $w$ integration in the Euclidean inversion formula. This contour deformation is only allowed if the integrand vanishes sufficiently fast for $w \to \infty$ (or equivalently, thanks to (2.6), for $w \to 0$). This means that the formula (2.12), as well as the original Caron-Huot formula [47], might fail for sufficiently low spin. While in the case without defect ref. [47] managed to derive a bound on the asymptotic behaviour of the correlator, no bound is currently known for the large $w$ behaviour of $F(r, w)$. Assuming that the correlator has a power-like behaviour for $w \to 0$, one has that if $F(r, w) \sim w^{-s_*}$ for $w \to 0$ then the formula is valid for $s > s_*$ [48]. The general expectation, based on the known examples, is that the correlator is indeed bounded by $w^{-s_*}$, with $s_*$ a small integer (no example is currently known where $s_* > 2$). We will find a similar limitation for the dispersion relation, but in this case, given a specific value of $s^*$, one can introduce an additional factor to improve the convergence of the formula (see Section 3).

A second inversion formula can be derived starting from the bulk partial wave decomposition [49]

$$F(z, \bar{z}) = \sum_{\ell} \int_{d/2-i\infty}^{d/2+i\infty} \frac{d\Delta}{2\pi i} c(\Delta, \ell) \Psi_{\Delta, \ell}(z, \bar{z}) \tag{2.14}$$

with

$$\Psi_{\Delta, \ell}(z, \bar{z}) = \frac{1}{2} \left( f_{\Delta, \ell}(z, \bar{z}) + \frac{K_{d-\Delta, \ell}}{K_{\Delta, \ell}} f_{d-\Delta, \ell}(z, \bar{z}) \right) \tag{2.15}$$

$$K_{\Delta, \ell} = \frac{\Gamma(\Delta - p - 1)\Gamma(\frac{\Delta-1}{2})}{\Gamma(\Delta - \frac{d}{2})\Gamma(\frac{\Delta-p-1}{2})} \kappa_{\Delta+\ell} \qquad \kappa_{\Delta+\ell} = \frac{\Gamma(\frac{\Delta+\ell}{2})^2}{2\pi^2 \Gamma(\Delta + \ell)\Gamma(\Delta + \ell - 1)} \tag{2.16}$$

In this case, the coefficient function $c(\Delta, \ell)$ contains all the information about the defect CFT data of the operators that are exchanged in the bulk OPE. Specifically, it has poles corresponding to their scaling dimensions and residues given by the product $c_{\phi\phi\mathcal{O}} a_{\mathcal{O}}$. Unlike the previous case, the relevant quantity to extract $c(\Delta, \ell)$ through a Lorentzian inversion formula is the double discontinuity defined by

$$\mathrm{dDisc} F(z, \bar{z}) = F(z, \bar{z}) - \frac{1}{2} F^{\circlearrowleft}(z, \bar{z}) - \frac{1}{2} F^{\circlearrowright}(z, \bar{z}) \tag{2.17}$$

where the functions $F^{\circlearrowleft}(z, \bar{z})$ and $F^{\circlearrowright}(z, \bar{z})$ are obtained by taking the analytic continuation around the point $\bar{z} = 0$ following an anticlockwise and a clockwise contour respectively. The final formula is very similar to the case without defect and, for identical scalars, it reads

$$c(\Delta, \ell) = (1 + (-1)^{\ell}) \frac{\kappa_{\Delta+\ell}}{2} \int_0^1 d^2z \, \mu(z, \bar{z}) \, f_{\ell+d-1, \Delta-d+1}(z, \bar{z}) \mathrm{dDisc} F(z, \bar{z}) \tag{2.18}$$

with

$$\mu(z, \bar{z}) = \left( \frac{(1-z)(1-\bar{z})}{\sqrt{z\bar{z}}} \right)^{\Delta} \frac{|z - \bar{z}|^{d-p-2} |1 - z\bar{z}|^p}{(1-z)^d (1-\bar{z})^d} \tag{2.19}$$

The physical content of the Lorentzian inversion formulae is that the discontinuity of the correlator around the point $\bar{z} = 1$ is sufficient to reconstruct the full defect spectrum, while

the double discontinuity around $\bar{z} = 0$ allows to reconstruct the bulk spectrum. Given the full spectrum of exchanged operators and their OPE coefficients, in principle one can resum the block expansions (2.2) and (2.8) to obtain the full correlator. Therefore, we conclude that the knowledge of either the discontinuity at $\bar{z} = 1$ or the double discontinuity at $z = 0$ is enough to obtain the full correlator. In practice, resumming the block expansion is usually a pretty hard task and for this reason it would be more convenient to have a formula that directly relates the discontinuity (or double discontinuity) to the full correlator. This is achieved through the defect dispersion relations.

## 3    Defect dispersion relations

The natural route one can follow to obtain a dispersion relation from the Lorentzian inversion formula is to insert the expression of the coefficient function $b(\hat{\Delta}, \ell)$ $(c(\Delta, \ell))$ into the partial wave decomposition (2.9) ( (2.14) ) and perform the integrals over $\hat{\Delta}$ and $s$ ($\Delta$ and $\ell$). For the case with the dDisc this seems to be the only viable way and we briefly discuss it in Section 3.2. For the case involving the single discontinuity at $\bar{z} = 1$, instead, there is an easier way to find a dispersion relation, which exploits the fact that the problem is effectively one-dimensional [5]. This allows to obtain the dispersion relation from a simple argument of complex analysis combined with the symmetry (2.6). In Appendix C we show that the formula we find here perfectly agrees with the one obtained through the Lorentzian inversion formula.

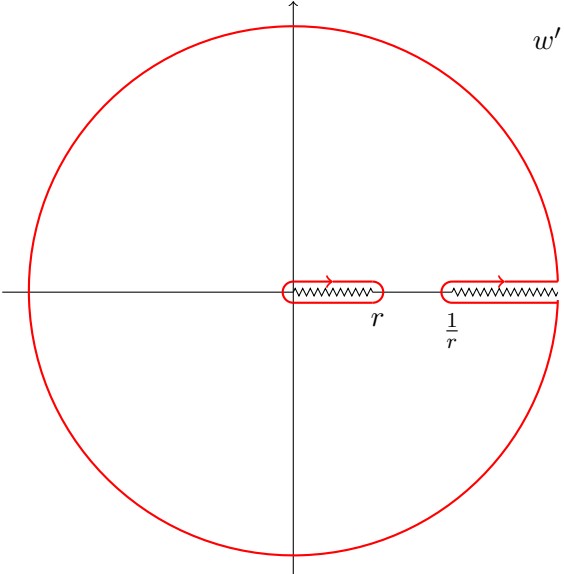

Figure 2: Contour deformation leading to the dispersion relation.

We start by observing that the radial part of the Lorentzian inversion formula (2.12) is essentially equivalent to its Euclidean counterpart, i.e. it does not involve any contour defor-

---

[5]We are grateful to M. Meineri for suggesting this strategy to us and for very helpful discussions on this point.

mation. As a consequence, one is led to expect that the associated dispersion relation only involves the angular variable $w$. Furthermore, since the inversion formula involves a single discontinuity it is clear that one should be able to find an elementary derivation of the dispersion relation. Indeed, fixing $r \in (0, 1)$, we see from the bulk and defect block expansions that the two-point function is regular everywhere in the complex $w$ plane except for two branch cuts at $(0, r)$ and $(\frac{1}{r}, \infty)$. Therefore we can simply use Cauchy's theorem for the variable $w$ and write

$$F(r, w) = \oint \frac{dw'}{2\pi i} \frac{F(r, w')}{w' - w} \tag{3.1}$$

We can now deform the contour to wrap it around the branch cuts as shown in Figure 2. We assume for the moment that we can drop the contribution from all the circles, i.e. the circle at infinity as well as the small circles around $w = 0, r, 1/r$ [6]. If this is case, we can write

$$F(r, w) = \int_0^r \frac{dw'}{2\pi i} \frac{1}{w' - w} \mathrm{Disc}_{0 < w' < r} F(r, w') + \int_{\frac{1}{r}}^\infty \frac{dw'}{2\pi i} \frac{1}{w' - w} \mathrm{Disc}_{w' > \frac{1}{r}} F(r, w') \tag{3.2}$$

Finally, we can use the symmetry $F(r, w) = F(r, \frac{1}{w})$ to change variable $w' \to \frac{1}{w'}$ in the second integral and get

$$F(r, w) = \int_0^r \frac{dw'}{2\pi i} \left( \frac{1}{w' - w} + \frac{1}{w' - \frac{1}{w}} - \frac{1}{w'} \right) \mathrm{Disc} F(r, w') \tag{3.3}$$

where the discontinuity is taken around the branch point at $w = r$ as in (2.11).

We now discuss the behaviour at $w' = 0$ and $w' = r$. The latter is controlled by the bulk OPE, i.e. the block expansion (2.8). The bulk blocks $f_{\Delta, \ell}(z, \bar{z})$ behave like $(w - r)^{\Delta - \ell}$ for $w \to r$ so the correlator is dominated by the exchanged operator with lowest twist, which is the identity. Then we conclude that for $w \to r$ the correlator goes like

$$F(r, w) \sim (w - r)^{-\Delta_\phi} \tag{3.4}$$

Therefore, for $\Delta_\phi > 1$, some care is needed in interpreting the formula (3.3). Since the integral we started from, before deforming the contour, was guaranteed to be finite, if no other singularity is present, also the contribution of the small circle around $w' = r$ combined with the integral in (3.3) must be finite. What happens is that the discontinuity in equation (3.3) must be interpreted in a distributional sense and values of $\Delta_\phi > 1$ give additional finite contributions localized at $w' = r$. We will see some examples in the explicit correlators we analyze below.

For the behaviour near $w = 0$, the situation is the same as in (2.12). While equation (2.12) allows to extract the defect CFT data for sufficiently high spin ($s > s_*$), the dispersion relation (3.3) reconstructs only part of the full correlator and the missing terms are given by low spin conformal blocks (summed over all the $\hat{\Delta}$) which are polynomials in $w$ but arbitrarily complicated functions of $r$. Therefore, it is important to improve this formula in order to

---

[6]Notice that, thanks to the symmetry (2.6), we just need to require a sufficiently good behaviour at $w = 0$ and $w = r$. This automatically implies that we can neglect the circles at $|w| = \infty$ and $w = \frac{1}{r}$

include the missing contributions. This can be easily done after knowing the behaviour of the correlator for $w \to 0$ (or equivalently for $|w| \to \infty$). In particular, if one knows that

$$F(r,w) \sim w^{-s_*} \quad \text{for} \quad w \to 0 \tag{3.5}$$

then we can define

$$\tilde{F}(r,w) = \left( \frac{r}{(w-r)(\frac{1}{w}-r)} \right)^{s_*+1} F(r,w) \tag{3.6}$$

which, by construction, goes like $w^{-1}$ at large $w$. Therefore, formula (3.3) certainly applies for the function $\tilde{F}$. We can then write down an improved version of the dispersion relation

$$\frac{F(r,w)}{(w-r)^{s_*+1}(\frac{1}{w}-r)^{s_*+1}} = \int_0^r \frac{dw'}{2\pi i} \left( \frac{1}{w'-w} + \frac{1}{w'-\frac{1}{w}} - \frac{1}{w'} \right) \text{Disc} \left[ \frac{F(r,w')}{(w'-r)^{s_*+1}(\frac{1}{w'}-r)^{s_*+1}} \right] \tag{3.7}$$

The main advantage of the formula (3.7) is that the discontinuity at $w = r$ is controlled by the bulk OPE. Indeed, the bulk blocks (A.9) are of the form

$$f_{\Delta,\ell} = (w-r)^{\frac{\Delta-\ell}{2}} \tilde{f}_{\Delta,\ell}(r,w) \tag{3.8}$$

where $\tilde{f}_{\Delta,\ell}(r,w)$ is regular at $w = r$. This turns out to be particularly powerful in a perturbative setting where the dimensions of the exchanged operators are deformed away from integer values $\Delta^{(0)}$

$$\Delta = \Delta^{(0)} + \epsilon\gamma^{(1)} + \epsilon^2\gamma^{(2)} + ...$$
$$\lambda_{\phi\phi\Delta} a_\Delta = \lambda^{(0)} + \epsilon\lambda^{(1)} + \epsilon^2\lambda^{(2)} + ... \tag{3.9}$$

Plugging this data in the bulk OPE expansion (2.8) we find the following structure

$$F(z,\bar{z}) = \sum \lambda^{(0)} f_{\Delta^{(0)},\ell} + \epsilon \left( \lambda^{(1)} f_{\Delta^{(0)},\ell} + \lambda^{(0)}\gamma^{(1)} \partial f_{\Delta^{(0)},\ell} \right)$$
$$+ \epsilon^2 \left( \lambda^{(2)} f_{\Delta^{(0)},\ell} + \left( \lambda^{(0)}\gamma^{(2)} + \lambda^{(1)}\gamma^{(1)} \right) \partial f_{\Delta^{(0)},\ell} + \frac{1}{2}\lambda^{(0)}(\gamma^{(1)})^2 \partial^2 f_{\Delta^{(0)},\ell} \right) + ... \tag{3.10}$$

where we used the notation $\partial f_{\Delta^{(0)},\ell} \equiv \partial_\Delta f_{\Delta,\ell}|_{\Delta=\Delta^{(0)}}$ and we suppressed the dependence of the blocks on the cross ratios. We can use this expansion to compute the discontinuity of the correlator, term by term in the expansion. The advantage of doing this is that not all of the above terms contribute. Indeed it is clear from the explicit form of the blocks (4.5) that the relevant discontinuity is given by the derivatives of the blocks, which produce logarithms when the derivative hits the exponent of $[(1-z)(1-\bar{z})]^{\frac{\Delta-\ell}{2}}$. We see that those terms at any given order depend on lower order OPE data and the anomalous dimensions at the order we are working at. Furthermore, since the anomalous dimension at a specific order multiplies the lowest-order OPE coefficient $\lambda^{(0)}$, we only need the anomalous dimensions of the operators that contribute at leading order. In other words, the discontinuity can be computed from a subset of OPE data. In particularly convenient cases, such as the one we are going to analyze in Section 5, this subset is finite and the formula provides a convenient and immediate way to reconstruct a correlator from a finite number of bulk CFT data. In general there may be

an additional source of discontinuities. If the correlator goes at large $w$ worse than $\frac{1}{w}$ we may need to introduce a prefactor, in order to drop the contribution at infinity in the dispersion relation. That prefactor may introduce additional poles in the OPE expansion, which will contribute to the discontinuity.

## 3.1 The contribution of bulk and defect identity operators

The simplest application we can consider is the bulk identity operator, i.e. the trivial defect. In the conventions of equation (2.1), the bulk two-point function without the defect corresponds to the function

$$F(z, \bar{z}) = \left( \frac{\sqrt{z\bar{z}}}{(1-z)(1-\bar{z})} \right)^{\Delta_\phi} = \left( \frac{r}{(1-rw)(1-\frac{r}{w})} \right)^{\Delta_\phi} \tag{3.11}$$

For $w \to 0$ the function (3.11) goes like $w^{\Delta_\phi}$ and for $w \to r$ it goes like $(w-r)^{-\Delta_\phi}$. Therefore, the circle contributions in Figure 2 can be neglected for $0 < \Delta_\phi < 1$ and in this regime we can simply use equation (3.3). The discontinuity of equation (3.11) at $w = r$ is given by

$$\text{Disc} \left( 1 - \frac{r}{w} \right)^{-\Delta_\phi} = 2i \sin(\pi \Delta_\phi) \left( \frac{r}{w} - 1 \right)^{-\Delta_\phi} \tag{3.12}$$

Then, one can check that

$$\frac{\sin(\pi \Delta_\phi)}{\pi} \int_0^r dw' \left( \frac{1}{w'-w} + \frac{1}{w'-\frac{1}{w}} - \frac{1}{w'} \right) \left( \frac{r}{(1-rw')(\frac{r}{w'}-1)} \right)^{\Delta_\phi} = \left( \frac{r}{(1-rw)(1-\frac{r}{w})} \right)^{\Delta_\phi} \tag{3.13}$$

for $0 < \Delta_\phi < 1$. For $\Delta_\phi > 1$ one can include the contributions of the small circle around $w = r$ and still find perfect agreement with the expectations.

A subtler example is the defect identity, i.e. $F(r, w) = a_\phi^2$ with $a_\phi$ the coefficient of the one-point function $\langle \phi \rangle$. In this case, we have $F(r, w) \sim w^0$ for $w \to 0$ so one needs to use equation (3.7) with $s_* = 0$. Naively, one would conclude that the discontinuity vanishes, however one should interpret equation (3.7) in a distributional sense such that

$$\text{Disc} \left( \frac{1}{w'-r} \right) = -2\pi i \, \delta(w'-r) \tag{3.14}$$

Using this relation, the dispersion relation is a trivial consequence of

$$-(w-r) \left( \frac{1}{w} - r \right) \int_0^r dw' \left( \frac{1}{w'-w} + \frac{1}{w'-\frac{1}{w}} - \frac{1}{w'} \right) \frac{1}{\frac{1}{w'}-r} \delta(w'-r) = 1 \tag{3.15}$$

Therefore, we checked that the dispersion relation correctly reproduces the disconnected part of a defect correlator, i.e. $F(r, w) = a_\phi^2 + \left( \frac{r}{(1-rw)(1-\frac{r}{w})} \right)^{\Delta_\phi}$. In section 5, we will apply our dispersion relation to more interesting examples and we will show that it allows to reproduce, in one line, perturbative results that were previously obtained by resumming the block expansion.

## 3.2 A dispersion relation with the double discontinuity

Plugging the result of the Lorentzian inversion formula (2.18) into the partial wave decomposition (2.14) and performing the sums over spin and dimension should lead to a dispersion relation involving the double discontinuity at $\bar{z} = 0$. In general, this is hard because no closed form is known for the bulk blocks. However, we should remember that this is true also for the case without the defect. To strategy of [47] was to derive a formula for $d = 4$ and $d = 2$ and then argue for its validity in general. We believe a similar logic applies to our case. In the particular case of $p = 2$, for any $d$, we can exploit the following fact

$$f_{\Delta,l}(z, \bar{z}) = \frac{(1 - z)(1 - \bar{z})}{1 - z\bar{z}} g^{d-2}_{\Delta-1,l+1}(1 - z, 1 - \bar{z}) \tag{3.16}$$

where $f_{\Delta,l}(z, \bar{z})$ is the bulk block and $g^d_{\Delta,l}(z, \bar{z})$ is the conformal block for a four-point function without the defect in dimension $d$. Then, if we rewrite the defect two-point function as

$$F(z, \bar{z}) = \left(\frac{\sqrt{z\bar{z}}}{(1 - z)(1 - \bar{z})}\right)^{\Delta_\phi} \frac{(1 - z)(1 - \bar{z})}{1 - z\bar{z}} \mathcal{G}(1 - z, 1 - \bar{z}) \tag{3.17}$$

then the function $\mathcal{G}(z, \bar{z})$ is a function which can be expanded in ordinary four-point function conformal blocks. This means that it can be computed it using the dispersion relation for Regge-bounded four point functions derived in [50]. The formula reads

$$
\begin{aligned}
\mathcal{G}(u, v) &= \mathcal{G}^t(u, v) + \mathcal{G}^u(u, v) \\
\frac{u}{v}\mathcal{G}^t(u, v) &= \int_0^1 d^2w \; K(u, v, u', v') \mathrm{dDisc}\left(\frac{u'}{v'}\mathcal{G}(u', v')\right)
\end{aligned}
\tag{3.18}
$$

where the u-channel expression is obtained by sending $z \to \frac{z}{z-1}$ and we have introduced the variables

$$u = z\bar{z} \qquad\qquad v = (1 - z)(1 - \bar{z}) \tag{3.19}$$

In these coordinates the kernel $K(u, v, u', v')$ is

$$K(u, v, u', v') = \frac{u - v + u' - v'}{64\pi(uvu'v')^{\frac{3}{4}}} x^{\frac{3}{2}} {}_2F_1\left(\frac{1}{2}, \frac{3}{2}, 2, 1 - x\right)(\theta(x - 1) - 4\delta(x - 1)) \tag{3.20}$$

where

$$x = \frac{16\sqrt{uvu'v'}}{[(\sqrt{u} + \sqrt{v})^2 - (\sqrt{u'} + \sqrt{v'})^2][(\sqrt{u} - \sqrt{v})^2 - (\sqrt{u'} - \sqrt{v'})^2]} \tag{3.21}$$

Therefore for our correlator we find

$$
\begin{aligned}
F^t(z, \bar{z}) &= \left(\frac{\sqrt{z\bar{z}}}{(1 - z)(1 - \bar{z})}\right)^{\Delta_\phi} \frac{z\bar{z}}{1 - z\bar{z}} \int_0^1 dwd\bar{w} \; K(1 - z, 1 - \bar{z}, 1 - w, 1 - \bar{w}) \\
&\quad \times \mathrm{dDisc}_{\bar{w}=0}\left[F(w, \bar{w})\left(\frac{\sqrt{w\bar{w}}}{(1 - w)(1 - \bar{w})}\right)^{-\Delta_\phi} \frac{1 - w\bar{w}}{w\bar{w}}\right]
\end{aligned}
\tag{3.22}
$$

This formula is derived and guaranteed to work for $p = 2$ and arbitrary $d$, however we notice that the function $F(z, \bar{z})$ has the same analytic structure for all $p$ and that the prefactors in the formula do not introduce new singularities. Since the original formula is derived from a contour deformation argument which is essentially based on the analytic structure of the functions, we conjecture that this formula works for all dimensions and codimensions, if $F(z, \bar{z})$ is appropriately bounded (otherwise we can simply add the suitable prefactors).

We checked this formula explicitly for the bulk identity correlator, which corresponds to

$$\mathcal{G}(1 - z, 1 - \bar{z}) = \frac{1 - z\bar{z}}{(1 - z)(1 - \bar{z})} \tag{3.23}$$

and the check reduces to the one performed in [50] for generalized free fields. We leave further checks of this relation for future work.

## 4 Dispersion relation for Boundary CFT

In this section we treat separately the case of codimension one, i.e. boundaries and interfaces. The main difference with the case of general defect lies in the presence of a single cross-ratio $z$, which is reflected in the absence of the transverse spin. In a sense, this provides a simplification. Nevertheless, it was observed in [22] that this feature leads to a different structure for the Lorentzian inversion formula, which contains two different contributions: a term involving a single discontinuity, controlled by the bulk OPE, and a term with a double discontinuity, controlled by the boundary OPE. Unlike the general defect case, it is not possible to find an inversion formula controlled by a single channel. Another drawback of this case is that the integration kernels are not known in a closed form, except when the difference of the external dimensions is an odd integer. The first obstacle is present also in our case. We can indeed use a simple contour deformation, as in section 3, to derive a dispersion relation involving two discontinuities, but, unlike the case of the general defect, there is no symmetry relating the two contributions. This leads to a less powerful formula, which requires knowledge of a subset of bulk CFT data as well as a subset of defect CFT data. Nevertheless, we will show in section 6 that, for some important perturbative settings, such as the boundary of the $O(N)$ model at the Wilson-Fisher fixed point, only a finite number of CFT data will be needed to reconstruct the full correlator.

We start from the two point function

$$\langle \phi(x_1)\phi(x_2) \rangle = \frac{F(z)}{(4|x_{1\perp}||x_{2\perp}|)^{\Delta_\phi}} \tag{4.1}$$

Notice that our conventions are slightly different compared to the general defect case to match those that are commonly used in the literature. The boundary block expansion reads

$$F(z) = \sum_{\hat{\Delta}} b_{\hat{\Delta}}^2 \, \hat{f}_{\hat{\Delta}}(z) \tag{4.2}$$

with

$$\hat{f}_{\hat{\Delta}}(z) = z^{\hat{\Delta}} \, {}_2F_1\left(\hat{\Delta}, \hat{\Delta} + 1 - \frac{d}{2}, 2\hat{\Delta} + 2 - d, z\right) \tag{4.3}$$

and the bulk block expansion is

$$F(z) = \left(\frac{z}{1-z}\right)^{\Delta_\phi} \sum_\Delta a_{\mathcal{O}} c_{\phi\phi\mathcal{O}} f_\Delta(z) \tag{4.4}$$

with

$$f_\Delta(z) = (1-z)^{\frac{\Delta}{2}} \, {}_2F_1\left(\frac{\Delta}{2}, \frac{\Delta}{2} + 1 - \frac{d}{2}, \Delta + 1 - \frac{d}{2}, 1 - z\right) \tag{4.5}$$

From the block expansions we see that the correlator has branch cuts for $z \in (-\infty, 0)$ and $z \in (1, +\infty)$.

Following the same logic as for the general defect we start from

$$F(z) = \frac{1}{2\pi i} \oint dz' \, \frac{F(z')}{z' - z} \tag{4.6}$$

where the integration contour encircle any regular point $z$.

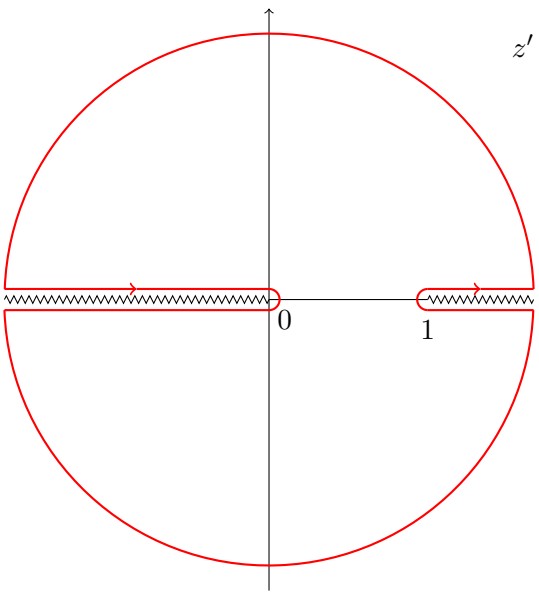

Figure 3: Contour deformation for the boundary case.

We can deform the contour as shown in Figure 3 and, assuming again that we can neglect the contribution from the circle at infinity and the two small circles at $z' = 0$ and $z' = 1$, we get

$$F(z) = -\frac{1}{2\pi i} \int_{-\infty}^0 dz' \, \frac{\mathrm{Disc}_{z'<0}(F(z'))}{z' - z} + \frac{1}{2\pi i} \int_1^\infty dz' \, \frac{\mathrm{Disc}_{z'>1}(F(z'))}{z' - z} \tag{4.7}$$

In other words, the two-point function can be computed from its discontinuity, defined as the difference between two analytic continuations of the correlator above and below a certain branch cut

$$\mathrm{Disc}(F(z)) = F(z + i\epsilon) - F(z - i\epsilon) \tag{4.8}$$

Before moving to the applications, let us discuss the convergence at $z' = 0, 1, \infty$. The lightest exchanged operator in the boundary channel is the boundary identity, which corresponds

to $\hat{\Delta} = 0$ in (4.2). All other boundary operators will contribute with terms that vanish for $z = 0$, so this limit does not present any problem. At $z = 1$, the leading contribution is given by the defect identity, which, using (4.4), corresponds to $F(z) \sim (1-z)^{-\Delta_\phi}$ exactly like the general defect for $w \to r$ (see (3.4)). Therefore, the same considerations that we applied in that case can be applied here.

The circle at $|z| \to \infty$ is again subtler as it is not directly controlled by an OPE expansion. Compared to the case of higher codimension, however, the situation is slightly different as the authors of [22] managed to derive a bound on the large $|z|$ behaviour of a boundary correlator, i.e.

$$F(z) \sim z^{\Delta_\phi} \quad \text{for} \quad |z| \to \infty \tag{4.9}$$

We can then improve the dispersion relation (4.7) by introducing a suitable prefactor as we did in (3.7). A possible choice is

$$F(z) = -\frac{z^{\Delta_\phi+1}}{2\pi i} \int_{-\infty}^{0} dz' \frac{\text{Disc}_{z'<0}\left(\frac{F(z')}{(z')^{\Delta_\phi+1}}\right)}{z'-z} + \frac{z^{\Delta_\phi+1}}{2\pi i} \int_{1}^{\infty} dz' \frac{\text{Disc}_{z'>1}\left(\frac{F(z')}{(z')^{\Delta_\phi+1}}\right)}{z'-z} \tag{4.10}$$

Of course this choice affects the behaviour of the integrand at $z' = 0$ and one should then reanalyse the contribution of the small circle around $z' = 0$ and take into account the contribution of the small circle around zero in case the integral diverges, as we discussed around (3.4). We stress that the choice (4.10) is by no means unique and we will see in Section 6 that in a perturbative setting, this freedom can be used to simplify the computation by reducing the number of exchanged operators that are needed to compute the discontinuities.

## 4.1 The contribution of bulk and boundary identity operators

As we did for the general defect, we can test our dispersion formula on the contribution of the identity operator. The identity operator in the boundary corresponds to $\hat{\Delta} = 0$, therefore, according to (4.3)

$$F(z) = a_\phi^2 \tag{4.11}$$

where $a_\phi$ is the coefficient of the one-point function $\langle \phi(x_\perp) \rangle = \frac{a_\phi}{|2x_\perp|^{\Delta_\phi}}$. If we want to reproduce this correlator from the dispersion relation, we need to improve its behaviour at large z by adding a prefactor of $\frac{1}{z}$. Then we have

$$\text{Disc}_{z<0}\left(\frac{1}{z}F(z)\right) = 2\pi i \delta(z) a_\phi^2$$

$$\text{Disc}_{z>1}\left(\frac{1}{z}F(z)\right) = 0 \tag{4.12}$$

and therefore we trivially have

$$F(z) = a_\phi^2 z \int_{-\infty}^{0} dz' \frac{1}{z'-z} \delta(z) = a_\phi^2 \tag{4.13}$$

From (4.4) we see that the bulk identity operator corresponds to a correlator

$$F(z) = \frac{z^{\Delta_\phi}}{(1-z)^{\Delta_\phi}} \tag{4.14}$$

Since $\Delta_\phi > 0$, it behaves like a constant for large $z$, therefore we need to add a prefactor in order to drop the large circle contributions in the contour integral. We have

$$\text{Disc}_{z<0}\left(\frac{1}{z}F(z)\right) = 2i\sin(\pi(\Delta_\phi - 1))\frac{(-z)^{\Delta_\phi}}{(1-z)^{\Delta_\phi}}$$

$$\text{Disc}_{z>1}\left(\frac{1}{z}F(z)\right) = -2i\sin(\pi(\Delta_\phi))\frac{z^{\Delta_\phi}}{(z-1)^{\Delta_\phi}} \tag{4.15}$$

and finally

$$
\begin{aligned}
F(z) &= -z\int_{-\infty}^{0}dz'\,\frac{1}{z'-z}\frac{\sin(\pi(\Delta_\phi-1))}{\pi}\frac{(-z)^{\Delta_\phi}}{(1-z)^{\Delta_\phi}}\\
&\quad -z\int_{1}^{\infty}dz'\,\frac{1}{z'-z}\frac{\sin(\pi\Delta_\phi)}{\pi}\frac{z^{\Delta_\phi}}{(z-1)^{\Delta_\phi}}\\
&= \frac{z^{\Delta_\phi}}{(1-z)^{\Delta_\phi}}
\end{aligned}
\tag{4.16}
$$

for $0 < \Delta_\phi < 1$. One can recover the $\Delta_\phi > 1$ result from analytic continuation or by reintroducing the contributions of small circles around $z = 0, 1$. When the external dimension is an integer, the discontinuity has to be interpreted as a delta function, as we said before.

## 5 Wilson line in $\mathcal{N} = 4$ SYM

A non-trivial example of conformal defect, which has been intensively analyzed in the literature, is the supersymmetric Wilson line in $\mathcal{N} = 4$ SYM theory

$$\mathcal{W} = \frac{1}{N}\text{Tr}\mathcal{P}\exp\left[\int d\tau\left(iA_\mu\dot{x}^\mu + |\dot{x}|\theta_I\phi^I\right)\right] \tag{5.1}$$

where we take the contour to be a straight line parametrized by $\tau$ and the unit vector $\theta^I$ ($\theta^I\theta_I = 1$) is a constant $SO(6)$ vector which we will take to be $\theta^6 = 1$ and $\theta^A = 0$ for $A = 1, \ldots, 5$. We are interested in the two-point function of $\frac{1}{2}$BPS operators of protected dimension $\Delta = P$, where $P$ is their R-charge. Let us start from the case $P = 2$

$$\mathcal{O}_{20'}^{IJ}(x) = \text{Tr}\left[\phi^I(x)\phi^J(x) - \frac{1}{6}\delta^{IJ}\phi^K(x)\phi_K(x)\right] \tag{5.2}$$

This operator is the superprimary of the $\mathcal{N} = 4$ stress tensor multiplet and it changes in the **20'** representation of the $SO(6)$ R-symmetry group[7]. To handle the dependence on R-symmetry indices it is often convenient to introduce the complex null vector $Y^I$, $Y^IY_I = 0$ and define

$$\mathcal{O}_2(x, y) = \text{Tr}\left(Y \cdot \phi(x)\right)^2 \tag{5.3}$$

By acting with a suitable differential operator it is always possible to transform functions of $Y$ into $SO(6)$ tensor structures. This expression readily generalizes for higher-dimensional BPS operators

$$\mathcal{O}_P(x, y) = \text{Tr}\left(Y \cdot \phi(x)\right)^P \tag{5.4}$$

---

[7]The $\frac{1}{2}$BPS operators in $\mathcal{N} = 4$ SYM are often identified by the Dynkin labels $[0, P, 0]$ and the operator (5.2) corresponds to the $[0, 2, 0]$ case.

The overall normalization of these operators is clearly related to the normalization of the field $\phi$ in the $\mathcal{N} = 4$ Lagrangian. In this work we take an abstract point of view, using only the symmetries and the internal consistency of the SCFT. We never need to make explicit reference to a Lagrangian formulation and all the quantities we compute are independent of the normalization of the field $\phi$. For this reason, we will take the operators (5.4) to be normalized as

$$\langle \mathcal{O}_P(Y_1, x_1) \mathcal{O}_P(Y_2, x_2) \rangle = \left( \frac{Y_1 \cdot Y_2}{x_{12}^2} \right)^P \tag{5.5}$$

We are interested in the two-point function in the presence of the Wilson line

$$\langle \mathcal{O}_P(Y_1, x_1) \mathcal{O}_P(Y_2, x_2) \rangle_{\mathcal{W}} = \frac{\langle \mathcal{O}_P(Y_1, x_1) \mathcal{O}_P(Y_2, x_2) \mathcal{W} \rangle}{\langle \mathcal{W} \rangle} \tag{5.6}$$

which can be written as [43, 44]

$$\langle \mathcal{O}_P(Y_1, x_1) \mathcal{O}_P(Y_2, x_2) \rangle_{\mathcal{W}} = \left( \frac{Y_1 \cdot \theta \; Y_2 \cdot \theta}{|x_{1\perp}||x_{2\perp}|} \right)^P \mathcal{F}_P(z, \bar{z}, \sigma) \tag{5.7}$$

where $z$ and $\bar{z}$ are the same kinematical cross-ratios as in (2.1) and $\sigma$ is the R-symmetry cross-ratio

$$\sigma = \frac{Y_1 \cdot Y_2}{Y_1 \cdot \theta \; Y_2 \cdot \theta} \tag{5.8}$$

It is not hard to realize that the function $\mathcal{F}_P(z, \bar{z}, \sigma)$ is a polynomial in $\sigma$ of order $P$. We can then write it as

$$\mathcal{F}_P(z, \bar{z}, \sigma) = \sum_{n=0}^{P} \sigma^{P-n} F_{P,n}(z, \bar{z}) \tag{5.9}$$

It is worth emphasizing that the dispersion relation (3.7) holds separately for the functions $F_{P,n}(z, \bar{z})$ provided that we know their behaviour at $w \to 0$. In the following, we will consider the expansion of the functions $F_{P,n}(z, \bar{z})$ at weak and strong coupling and discuss how a subset of the bulk operators is sufficient to reconstruct the full correlator. Before doing this, we need to introduce the selection rules and the superconformal block decompositions in the bulk channel.

## 5.1 Selection rules and superconformal block expansion

The function $\mathcal{F}_P(z, \bar{z}, \sigma)$ can be expanded in the bulk and in the defect channel. The block expansions discussed in Section (3) are improved to superblock expansions in the presence of supersymmetry. Here we summarize the results for the selection rules and the superblocks derived in [43, 44] and we refer the reader to these works for a thorough discussion.

The operators $\mathcal{O}_P$ are superprimaries of the $\frac{1}{2}$BPS multiplet $\mathcal{B}_{[0,P,0]}$ (here we used the notation of [55] for the $\mathcal{N} = 4$ supermultiplets). In the bulk channel, we are interested in taking the fusion $\mathcal{B}_{[0,P,0]} \times \mathcal{B}_{[0,P,0]}$ and select the supermultiplets with a non-vanishing defect one-point function. Combining the results of [56] and [31, 44], we have

$$\mathcal{B}_{[0,P,0]} \otimes \mathcal{B}_{[0,P,0]} \to \mathbb{1} \oplus \sum_{k=1}^{P} \mathcal{B}_{[0,2k,0]} \oplus \sum_{k=1}^{P-1} \sum_{\ell} \mathcal{C}_{[0,2k,0],\ell} \oplus \sum_{k=0}^{P-2} \sum_{\Delta,\ell} \mathcal{A}_{[0,2k,0],\ell}^{\Delta} \tag{5.10}$$

where $\mathcal{C}_{[0,2k,0],\ell}$ are semishort multiplets with protected scaling dimension $\Delta = 2+2k+\ell$, while $\mathcal{A}^{\Delta}_{[0,2k,0],\ell}$ are long multiplets with arbitrary scaling dimension $\Delta > 2+2k+\ell$. Each exchanged supermultiplet corresponds to a superblock $\mathcal{F}_{\Delta,\ell}(z,\bar{z},\sigma)$, which encodes the contributions of all the superdescendants in the associated supermultiplet and for this reason it can be expressed as a linear combination of ordinary bulk blocks. Then the correlator can be expanded as

$$\mathcal{F}_P(z,\bar{z},\sigma) = \left(\frac{\sqrt{z\bar{z}}\,\sigma}{(1-z)(1-\bar{z})}\right)^P \sum_{\mathcal{O}} \lambda_{PP\mathcal{O}} a_{\mathcal{O}} \mathcal{F}_{\Delta,\ell}(z,\bar{z},\sigma) \qquad (5.11)$$

where the sum is taken over the superprimary operators $\mathcal{O}$ for each of the supermultiplets that are allowed to appear in the OPE (5.10). The coefficients are given by a product of a bulk three-point function $\lambda_{\mathcal{O}}$ and a one-point function $a_{\mathcal{O}}$. When also the exchanged multiplet is short, the OPE coefficients are known exactly

$$\begin{aligned} \lambda_{P_1 P_2 P_3} &= \frac{\sqrt{P_1 P_2 P_3}}{N} \\ a_P &= \frac{\sqrt{\lambda P}}{2^{\frac{P}{2}+1} N} \frac{I_P(\sqrt{\lambda})}{I_1(\sqrt{\lambda})} \end{aligned} \qquad (5.12)$$

Each superconformal block $\mathcal{F}_{\Delta,\ell}(z,\bar{z},\sigma)$ can be expressed as a linear combination of ordinary bulk conformal blocks $f_{\Delta,\ell}(z,\bar{z})$ (A.9) as shown in Appendix B. In the defect channel, each supermultiplet $\mathcal{B}_{[0,P,0]}$ is expanded in defect superblocks. For this work, we will not need the details of this expansion, which can be found in [43, 44]. Our next goal is to understand the implications of the dispersion relation (3.7) for this setup.

## 5.2 Strong coupling

We start our analysis at large $N$ and large $\lambda$, where the supersymmetric Wilson loop is described as the string worlsheet of minimal area ending on the contour of the loop. The correlator of bulk operators can be computed perturbatively by Witten diagrams where some propagators end on the worldsheet. At large $N$ and large $\lambda$, the leading contribution to the correlator $\langle \mathcal{O}_P \mathcal{O}_P \rangle_{\mathcal{W}}$ is just (5.7), i.e. the two-point function without the defect. At order $1/N^2$ we have two contributions: a disconnected one, which simply gives the product of two one-point functions at leading order and a second one, suppressed by $\frac{1}{\sqrt{\lambda}}$ which is the leading connected contribution, i.e.

$$\langle \mathcal{O}_P \mathcal{O}_P \rangle_{\mathcal{W}} = \langle \mathcal{O}_P \mathcal{O}_P \rangle + \frac{\lambda}{N^2} \left( \langle \mathcal{O}_P \rangle_{\mathcal{W}}^{(0)} \langle \mathcal{O}_P \rangle_{\mathcal{W}}^{(0)} + \frac{1}{\sqrt{\lambda}} \langle \mathcal{O}_P \mathcal{O}_P \rangle_{\mathcal{W}}^{(1)} + O(1/\lambda) \right) + O\left(\frac{1}{N^4}\right) \qquad (5.13)$$

Correspondingly, the function $\mathcal{F}_P(z,\bar{z},\sigma)$ reads

$$\mathcal{F}_P(z,\bar{z},\sigma) = \left(\frac{\sqrt{z\bar{z}}\,\sigma}{(1-z)(1-\bar{z})}\right)^P + \frac{\lambda}{N^2} \left( \frac{P}{2^{P+2}} + \frac{1}{\sqrt{\lambda}} \mathcal{F}_P^{(1)}(z,\bar{z},\sigma) + O(1/\lambda) \right) + O\left(\frac{1}{N^4}\right) \qquad (5.14)$$

where we used the result for the one-point function (5.12). The function $\mathcal{F}_P^{(1)}(z,\bar{z},\sigma)$ has been computed for $P = 2, 3, 4$ in [44] by extracting the defect CFT data using the Lorentzian

inversion formula and resumming the defect block expansion. Here, using the dispersion relation (C.9) we can skip the intermediate step and recover the result by performing a very simple integration. This gives a clear understanding of the reason why the final result for the correlator is particularly simple.

The crucial observation of [44] is that very few operators contribute to the discontinuity at $\bar{z} = 1$. In particular, at large $N$ and large $\lambda$, the bulk theory is described by an effective supergravity theory in $AdS_5$ and the spectrum of light excitations contains only the protected Kaluza-Klein modes in the short $\mathcal{B}_{[0,P,0]}$ multiplets and double trace operators with dimension

$$\Delta = 2P + 2n + \ell + O(N^{-2}) \tag{5.15}$$

Notice that the twist of the double trace operators ($\tau = \Delta - \ell = 2P + 2n$) is significantly higher than the lower bound allowed by the selection rules (5.10) which would allow for long operators of twist as low as two. Then the factor $(w-r)^{\frac{\Delta-\ell}{2}}$ in (3.8) together with the prefactor in (5.11) ensures that the contribution of the double trace operators goes like $(w - r)^n$ with $n \geq 0$. Therefore, double trace will not contribute to the discontinuity as long as no improvement is needed, i.e. if we can use (3.3) instead of (3.7). In [44] it was argued that the behaviour of the functions $F_{P,n}^{(1)}(z, \bar{z})$ in (5.9) for $w \to 0$ is $F_{P,n}^{(1)}(r, w) \sim w^{P-n-1}$ so that the improvement is needed only for $F_{P,P-1}^{(1)}$ and $F_{P,P}^{(1)}$. For all the cases when the improvement is not needed, only short operators contribute to the discontinuity.

For $P = 2$, no improvement is needed to compute $F_{2,0}^{(1)}(z, \bar{z})$. Following [44], we compute the discontinuity from the OPE, keeping only the negative powers in the superblock expansion. This gives a $\delta$-function contribution

$$\mathrm{Disc}(F_{2,0}^{(1)}(r, w)) = -2\pi i \lambda_{222}^{(1)} \delta(r - w) \frac{\left(r^2 w \left(r^4 - 2r^2 \log\left(r^2\right) - 1\right)\right)}{\left(r^2 - 1\right)^3 (rw - 1)} \tag{5.16}$$

where $\lambda_{222}^{(1)}$ is the strong coupling expansion of (5.12). More generally, we define

$$\lambda_{PP\mathcal{O}} a_{\mathcal{O}} = \frac{\sqrt{\lambda}}{N^2} \lambda_{PP\mathcal{O}}^{(1)} + O(\lambda) \tag{5.17}$$

Equation (5.16) can be immediately integrated in (3.3) obtaining with no effort the final result of [44]

$$F_{2,0}^{(1)}(r, w) = -\lambda_{222}^{(1)} \frac{r^2 w \left(r^4 - 2r^2 \log\left(r^2\right) - 1\right)}{\left(r^2 - 1\right)^3 (r - w)(rw - 1)} \tag{5.18}$$

In principle, we could follow a similar procedure for the other R-symmetry components, but the situation is more complicated. For example, the function $F_{P,1}^{(1)}(r, w)$ goes like $w^0$ for $w \to 0$ and it needs an improvement given by (3.7) with $s_* = 0$. Therefore, it will receive contributions also from twist-four double traces with any spin. This is still a simplification compared to resumming the full block expansion as it requires to resum a single infinite family of operators (fixed twist, any spin), instead of two. Still, in this case this procedure is not very practical and a more efficient one, proposed in [44], would be to use the dispersion relation (3.3) to determine $F_{P,1}^{(1)}(r, w)$ up to low spin ambiguities and then fix the ambiguities using superconformal Ward identities.

In general, we can carry out this same procedure for all $\langle \mathcal{O}_P \mathcal{O}_P \rangle$, confirming and extending the results of [44]. Indeed, in this holographic setup only short operators contribute to the discontinuity in all the cases where we do not need to improve the dispersion relation. This means that in (5.10) only the multiplets $\mathcal{B}_{[0,2k,0]}$ need to be considered for the discontinuity. Furthermore the multiplet $\mathcal{B}_{[0,2P,0]}$ is too high in dimension and it does not contribute. We can then compute the discontinuity at the order we are interested in as

$$\text{Disc}(\mathcal{F}_P^{(1)}(z,\bar{z},\sigma)) = \text{Disc}\left[ \left( \frac{\sqrt{z\bar{z}}\sigma}{(1-z)(1-\bar{z})} \right)^P \sum_{k=1}^{P-1} \lambda_{PP2k} a_{2k} \mathcal{G}_{[0,2k,0]}(z,\bar{z},\sigma) \right] \tag{5.19}$$

and use the dispersion formula to find the $F_{P,p}^{(1)}$ up to $p = P - 2$. Notice that, as for the case $P = 2$ all the contributions are distributions, specifically $\delta$-functions or derivatives thereof. For example for $P = 3$

$$\text{Disc}(F_{3,0}^{(1)}(r,w)) = \lambda_{334}^{(1)} \delta(w-r) \frac{\left( r^3 w \left( r^6 + 9r^4 - 9r^2 - 6 \left( r^4 + r^2 \right) \log\left( r^2 \right) - 1 \right) \right)}{4 \left( r^2 - 1 \right)^5 (rw - 1)}$$

$$- \frac{\lambda_{332}^{(1)} \, \delta'(w-r) \, r^3 w}{4 \left( r^2 - 1 \right)^5 (rw - 1)^2} \left( 2r^2 \left( -5r^4 w + 3r^3 \left( w^2 + 1 \right) - 2r^2 w + 3r \left( w^2 + 1 \right) - 5w \right) \log\left( r^2 \right) \right.$$

$$\left. + \left( r^2 - 1 \right) \left( 3r^6 w - r^5 \left( w^2 + 1 \right) + 9r^4 w - 10r^3 \left( w^2 + 1 \right) + 9r^2 w - r \left( w^2 + 1 \right) + 3w \right) \right) \tag{5.20}$$

Which gives the expected result [44]

$$F_{3,0}^{(1)}(r,w) = -\frac{3}{4} \frac{r^3 w^2 \left( r^4 - 4r^2 \log(r) - 1 \right)}{(r^2 - 1)^3 (r-w)^2 (rw - 1)^2} \tag{5.21}$$

Computing at higher $P$ does not involve any conceptual obstacle and it is only an algorthmic procedure. We checked the conjecture of [44] for the function $F_{P,0}^{(1)}$

$$F_{P,0}^{(1)}(z,\bar{z}) = -\frac{P}{4} \frac{(z\bar{z})^{\frac{P}{2}}}{[(1-z)(1-\bar{z})]^{P-1}} \left[ \frac{1+z\bar{z}}{(1-z\bar{z})^2} + \frac{2z\bar{z}\log(z\bar{z})}{(1-z\bar{z})^3} \right] \tag{5.22}$$

up to very high values of $P$ and we always found perfect agreement. We also derived the functions $F_{P,p}^{(1)}(z,\bar{z})$ with $p \le P - 2$ for several values of $P$. As an example, in Appendix D we spell out the results for $F_{5,p}^{(1)}$, which is the first case that did not appear in [44].

## 5.3 Weak coupling

At weak coupling things get more complicated as there is no simplification in the spectrum of exchanged operators and all the operators that are allowed to appear actually appear. Still, we can use some of the results obtained in [43] for $\langle \mathcal{O}_2 \mathcal{O}_2 \rangle$ to explore the implications of the dispersion relation. The function $\mathcal{F}_2(z,\bar{z},\sigma)$ is expanded at small $\lambda$ as

$$\mathcal{F}_2(z,\bar{z},\sigma) = \mathcal{F}_2^{(0)}(z,\bar{z},\sigma) + \frac{\lambda}{N^2} \mathcal{F}_2^{(1)}(z,\bar{z},\sigma) + \frac{\lambda^2}{N^2} \mathcal{F}_2^{(2)}(z,\bar{z},\sigma) \tag{5.23}$$

and, at each order the functions $\mathcal{F}_2^{(i)}$ are decomposed as in (5.9). At order zero there is a disconnected contribution, corresponding to the identity operator being exchanged in the bulk. The only non-vanishing function is then (in our normalization)

$$F_{2,0}^{(0)}(r,w) = 1 \tag{5.24}$$

At order one, if we look at the superblock boundary expansion, according to the argument below (3.10), there are no logarithmic contributions from the anomalous dimensions, since at this order they would be proportional to the order zero coefficients. The discontinuity is then generated by the negative powers in the OPE that come only from the short multiplets. Indeed if we expand around $w = r$ we have only one divergent contribution for the function $F_{2,1}^{(1)}$

$$F_{2,1}^{(1)} = \frac{4\lambda_{222}a_2^{(1)}\, r}{(1-wr)(1-\frac{r}{w})} + \text{regular} \tag{5.25}$$

where $a_2^{(1)}$ is the leading order term in the weak coupling expansion of (5.12). From this we find

$$\text{Disc}(F_{2,1}^{(1)}) = 2\pi i \frac{4\lambda_{222}a_2^{(1)}\, rw}{(1-rw)}\delta(w-r) \tag{5.26}$$

Inserting this into (3.3) we find

$$F_{2,1}^{(1)} = \frac{r}{(1-wr)(1-\frac{r}{w})} \tag{5.27}$$

in agreement with [43] At second order things become more complicated. For the first time we get terms with logarithms in the OPE expansion (3.10). It turns out that these terms only contribute to $F_{2,2}^{(2)}$ and $F_{2,1}^{(2)}$. We can write down these terms as infinite sums but we cannot evaluate them because the bulk blocks are not known in closed form.

The $F_{2,0}^{(2)}$ term is simpler, and its discontinuity receives contributions only from negative powers. Assuming that $F_{2,0}^{(2)}$ goes as a constant for $w$ going to infinity, we find

$$
\begin{aligned}
\text{Disc}(F_{2,0}^{(2)}) = {} & \frac{rw}{1-rw}\delta(w-r)\Big(\lambda_{224}a_4^{(2)}\frac{3r^2\left(-2r^2+\left(r^2+1\right)\log\left(r^2\right)+2\right)}{(r^2-1)^3} \\
& - \sum_{\ell=0}^{\infty}\lambda_{22\ell}a_\ell^{(2)}\frac{(\ell+1)r^2\left(1-r^2\right)^{\ell+2}\,_2F_1\left(\ell+4,\ell+4;2\ell+8;1-r^2\right)}{16(\ell+2)}\Big)
\end{aligned}
\tag{5.28}
$$

where $\lambda_{22\ell}a_\ell^{(2)}$ is the product of OPE coefficients and one-point function of twist-four semishort operators in the $\mathcal{C}_{[0,4,0],\ell}$ multiplet. From (3.3) we get

$$
\begin{aligned}
F_{2,0}^{(2)} = {} & \lambda_{224}a_4^{(2)}\frac{3r^2\left(-2r^2+\left(r^2+1\right)\log\left(r^2\right)+2\right)}{(r^2-1)^3} \\
& - \sum_{\ell=0}^{\infty}\lambda_{22\ell}a_\ell^{(2)}\frac{(\ell+1)r^2\left(1-r^2\right)^{\ell+2}\,_2F_1\left(\ell+4,\ell+4;2\ell+8;1-r^2\right)}{16(\ell+2)}
\end{aligned}
\tag{5.29}
$$

We see that $F_{2,0}^{(2)}$ can be reconstructed only from short and twist-four semi-short operators. Since there is no closed form for the semi-short OPE coefficients, we could not perform this sum analytically. However we have checked that this expansion reproduces the known result [43], which we reproduce here for convenience

$$
\begin{aligned}
F_{2,0}^{(2)} &= \frac{1}{128\pi^2}\Big(2\mathrm{Li}_2\left(r^2\right) + 4\mathrm{Li}_2\left(\frac{1+r}{2}\right) - 4\mathrm{Li}_2\left(\frac{1-r}{2}\right) - 8\mathrm{Li}_2(r) - 4\log(1-r)\log(r) \\
&\quad + 4\log\left(\frac{(1-r)r}{r+1}\right)\log(r+1) + 8\log(2)\tanh^{-1}(r) + \pi^2\Big)
\end{aligned}
$$

(5.30)

# 6 The boundary in the O(N) critical model

The advantage of the dispersion relation (4.7) is that, in some cases, the discontinuity can be computed without knowing the full correlator. One example where this is possible is the $O(N)$ statistical model at the Wilson-Fisher fixed point. We start from a free scalar theory in the presence of a boundary. This was analyzed with bootstrap techniques in [17] where it was found that two possible solutions to the boundary crossing equation are of the form

$$
F^N(z) = \left(\frac{z}{1-z}\right)^{\Delta_\phi} + z^{\Delta_\phi} \qquad\qquad F^D(z) = \left(\frac{z}{1-z}\right)^{\Delta_\phi} - z^{\Delta_\phi} \qquad (6.1)
$$

with $\Delta_\phi = \frac{d}{2} - 1$. These two solutions are associated to Neumann and Dirichlet boundary conditions respectively. In this case, the expansion in bulk blocks features a single primary operator, $\phi^2$ with dimension $\Delta = 2\Delta_\phi$, which is the only exchanged operator, other than the identity, that has a non-vanishing one-point function (whose sign changes from Neumann to Dirichlet). Also in the boundary channel expansion a single operator is exchanged: $\hat{\phi}$ of dimension $\hat{\Delta} = \Delta_\phi$ for Neumann (when $\partial_\perp \hat{\phi} = 0$) and $\partial_\perp \hat{\phi}$ of dimension $\hat{\Delta} = \Delta_\phi + 1$ for Dirichlet (when $\hat{\phi} = 0$) [8]. We start from this solution and we perturb it by expanding the bulk and defect CFT data. Let us start from the bulk channel expansion (4.4) and, after defining $a \equiv a_{\mathcal{O}}c_{\phi\phi\mathcal{O}}$, we take

$$
\begin{aligned}
\Delta &= \Delta^{(0)} + \epsilon\gamma^{(1)} + \epsilon^2\gamma^{(2)} + \dots \\
a &= a^{(0)} + \epsilon a^{(1)} + \epsilon^2 a^{(2)} + \dots
\end{aligned}
$$

(6.2)

Plugging this data in the bulk OPE expansion, we have the following structure

$$
\begin{aligned}
\left(\frac{1-z}{z}\right)^{\Delta_\phi} F(z) &= \sum a^{(0)}f_{\Delta^{(0)}}(z) + \epsilon\left(a^{(1)}f_{\Delta^{(0)}}(z) + a^{(0)}\gamma^{(1)}\partial_\Delta f_\Delta(z)|_{\Delta^{(0)}}\right) \\
&+ \epsilon^2\left(a^{(2)}f_{\Delta^{(0)}}(z) + (a^{(0)}\gamma^{(2)} + a^{(1)}\gamma^{(1)})\partial_\Delta f_\Delta(z)|_{\Delta^{(0)}} + \tfrac{1}{2}a^{(0)}(\gamma^{(1)})^2\partial_\Delta^2 f_\Delta(z)|_{\Delta^{(0)}}\right) + \dots
\end{aligned}
$$

(6.3)

where the sum is taken over the exchanged bulk operators. In the case we are discussing, there is a single bulk operator with non-vanishing $a^{(0)}$ and its anomalous dimension will appear in

---

[8]The derivatives $\partial_\perp$ refer to derivatives in the direction orthogonal to the boundary and therefore $\partial_\perp\hat{\phi}$ is not a boundary descendant.

the combination $a^{(0)}\gamma^{(1)}$ at order $\epsilon$. All the operators with a one-point function of order $\epsilon$, i.e. non-vanishing $a^{(1)}$, will enter at first order with their classical dimension $\Delta^{(0)}$.

An analogous expansion holds in the defect channel

$$\hat{\Delta} = \hat{\Delta}^{(0)} + \epsilon\hat{\gamma}^{(1)} + \epsilon^2\hat{\gamma}^{(2)} + ...$$
$$b_{\hat{\Delta}}^2 = b^{(0)} + \epsilon b^{(1)} + \epsilon^2 b^{(2)} + ...$$

(6.4)

and inserting these into (4.2) we obtain

$$F(z) = \sum b^{(0)}\hat{f}_{\hat{\Delta}^{(0)}}(z) + \epsilon\left(b^{(1)}\hat{f}_{\hat{\Delta}^{(0)}}(z) + b^{(0)}\hat{\gamma}^{(1)}\partial_{\hat{\Delta}}\hat{f}_{\hat{\Delta}}(z)|_{\hat{\Delta}^{(0)}}\right)$$
$$+ \epsilon^2\left(b^{(2)}\hat{f}_{\hat{\Delta}^{(0)}}(z) + (b^{(0)}\hat{\gamma}^{(2)} + b^{(1)}\hat{\gamma}^{(1)})\partial_{\hat{\Delta}}\hat{f}_{\hat{\Delta}}(z)|_{\hat{\Delta}^{(0)}} + \tfrac{1}{2}b^{(0)}(\hat{\gamma}^{(1)})^2\partial_{\hat{\Delta}}^2\hat{f}_{\hat{\Delta}}(z)|_{\hat{\Delta}^{(0)}}\right) + ...$$

(6.5)

Also in this case, the only exchanged operator at leading order will appear at order $\epsilon$ with its anomalous dimension $\hat{\gamma}^{(1)}$. All other operators with non-vanishing $b^{(1)}$ will enter with their classical dimension.

We would like to use these expansions to compute the discontinuities of the correlator by exchanging the discontinuity and the sum. In other words, we compute the discontinuity of each term in the above sum first, and then we sum the contributions. The advantage of doing this is that, thanks to the perturbative expansion, not all of the above terms are necessary to compute the discontinuity. Let us see how this happens. The idea is to use the boundary block expansion to compute the discontinuity at $z' = 0$ in (4.7) and the bulk block expansion for the discontinuity at $z' = 1$. If we consider the boundary OPE expansion, it is clear from the explicit expression of the boundary blocks (4.3) that the discontinuity at $z = 0$ is given by two sources. The first source are the logarithms from the terms $\partial_\Delta f_\Delta(z)$, which contain $\partial_\Delta z^\Delta$. Notice that these terms are proportional to the anomalous dimensions at the order we are working and to lower order OPE coefficients. Therefore they are absent for all the operators that did not appear in the previous order. The other possible discontinuities are poles that come from the additional factor we need to add in (4.10) to improve the convergence at infinity. Indeed, we should always keep in mind that the discontinuity must be interpreted in a distributional sense. Therefore, if the function $F(z)$ has a pole at $z = 0$, its discontinuity will be given by

$$\text{Disc}_{z<0}(z^{-n}) = \frac{2\pi i(-1)^n\partial^{n-1}(\delta(z))}{(n-1)!}$$

(6.6)

The important point is that using (6.5) we can compute the discontinuity of a correlator from a subset of the defect CFT data. This remains true also at higher orders and it works also for the $z = 1$ discontinuity, where we can use the bulk expansion. One drawback of this procedure is that we need to know the anomalous dimensions of some operators at the order we are interested in, i.e. the discontinuity is not completely fixed at a given order by lower order data, as it is normally the case for the double discontinuity. Nevertheless, for the case at end we will see that this procedure is still extremely powerful as we only need a finite number of defect CFT data to reconstruct the full correlator up to order $\epsilon^2$.

Before showing how to obtain the result, let us comment on the relation between our work and [20], where the authors also used the discontinuity to bootstrap the results that we

are reproducing here [9]. In their case, they took one particular discontinuity of the crossing equation, which allowed them to extract the boundary CFT data using only consistency of the crossing equation. Here we use two different discontinuities to reconstruct the correlator (which they computed by resumming the OPE expansion) and a finite number of boundary OPE data are the input for our formula. In this sense the two approaches are complementary. Notice also that we always use the OPE expansions in their region of convergence so we do not have to worry the subtleties that were discussed in [20].

## 6.1  Order $\epsilon$

Let us start from order $\epsilon$. It was pointed out in [17] that it is sufficient to add a bulk block corresponding to the operator $\phi^4$ with classical dimension $\Delta_{\phi^4} = 2d - 4$ to find a consistent solution to the defect crossing equation. Given the reduced number of operators that are relevant for this order, the dispersion relation is not more convenient than the ordinary block expansion in this case. Nevertheless, we briefly show how it works as it helps in illustrating the main features that will be useful at higher orders. From now on, we will focus on the Neumann case although everything works exactly the same way for Dirichlet. First of all, let us take $d = 4 - \epsilon$ and define a rescaled correlator

$$\tilde{F}(z) = \frac{1}{z(1-z)} F(z) \tag{6.7}$$

Consider first the discontinuity at $z = 0$, which is controlled by the boundary expansion (6.5) with a single operator $\hat{\phi}$ of dimension

$$\hat{\Delta}_{\hat{\phi}} = \frac{d}{2} - 1 + \epsilon \gamma_{\hat{\phi}}^{(1)} + O(\epsilon^2) = 1 + \epsilon(\hat{\gamma}_{\hat{\phi}}^{(1)} - 1/2) + O(\epsilon^2) \tag{6.8}$$

The order $\epsilon$ term for $\tilde{F}$ reads

$$\tilde{F}^{(1)}(z) = \frac{b_{\hat{\phi}}^{(1)} \hat{f}_1(z) + b^{(0)}(\hat{\gamma}_{\hat{\phi}}^{(1)} - 1/2) \partial_{\hat{\Delta}} \hat{f}_{\hat{\Delta}}(z)|_{\hat{\Delta}=1}}{z(1-z)} \tag{6.9}$$

In this case the block expansion coincides with the full correlator, but the idea is that, in general, we always extract the discontinuity from the expansion without knowing the full correlator. Since $\hat{f}_1(z) = z(1 + \frac{1}{1-z})$, the only contribution to the discontinuity comes from the derivative $\partial_{\hat{\Delta}} \hat{f}_{\hat{\Delta}}(z)|_{\hat{\Delta}=1}$ and specifically from the logarithm that is generated by the action of the derivative on the term $z^{\hat{\Delta}}$ in (4.3). Thus we have

$$\mathrm{Disc}_{z<0} \tilde{F}^{(1)}(z) = 2\pi i \, b_{\hat{\phi}}^{(0)}(\hat{\gamma}_{\hat{\phi}}^{(1)} - 1/2) \left( \frac{1}{1-z} + \frac{1}{(1-z)^2} \right) \tag{6.10}$$

A parallel argument allows to derive the discontinuity at $z = 1$ using the bulk expansion (6.3) with two exchanged operators

$$\Delta_{\phi^2} = d - 2 + \epsilon \gamma_{\phi^2}^{(1)} + O(\epsilon^2) = 2 + \epsilon(\gamma_{\phi^2}^{(1)} - 1) + O(\epsilon^2) \qquad a_{\phi^2} = 1 + \epsilon a_{\phi^2}^{(1)} + O(\epsilon^2) \tag{6.11}$$

$$\Delta_{\phi^4} = 4 + O(\epsilon) \qquad\qquad\qquad\qquad\qquad\qquad a_{\phi^4} = \epsilon a_{\phi^4}^{(1)} + O(\epsilon^2) \tag{6.12}$$

---

[9]The same result was also rederived in [57] by mapping the problem to AdS and solving the bulk equations of motion

where for the operator $\phi^4$ we neglected the anomalous dimension because its OPE coefficient $a_{\phi^4}$ is already of order $\epsilon$. For this channel we also need to take into account the prefactor $\left(\frac{z}{1-z}\right)^{\Delta_\phi}$ in equation (6.3) with

$$\Delta_\phi = \frac{d}{2} - 1 + \epsilon\gamma_\phi^{(1)} + O(\epsilon^2) = 1 + \epsilon(\gamma_\phi^{(1)} - 1/2) \tag{6.13}$$

All in all we get

$$\tilde{F}^{(1)}(z) = \frac{a_{\phi^2}^{(1)} f_2(z) + a_{\phi^4}^{(1)} f_4(z) + a_{\phi^2}^{(0)}(\gamma_{\phi^2}^{(1)} - 1)\partial_\Delta f_\Delta(z)|_{\Delta=2} + \log\frac{1-z}{z}(\gamma_\phi^{(1)} - 1/2)f_2(z)}{(1-z)^2} \tag{6.14}$$

Here all the terms contribute to the discontinuity: some of them contribute with a $\delta$-function and some with a logarithm. The only term which requires additional care is the one that comes from the $\log(1 - z)$ in the last term of (6.14). In that case we have $\mathrm{Disc}\left(\frac{\log(1-z)}{1-z}\right)$. The obvious way to do the computation is to compute explicitly the contribution from the little circle around $z = 1$ in the contour in Figure 3. However it turns out one can give to this discontinuity a sense in terms of a distribution, i.e. given a test function $f(x)$, the discontinuity for $x < 0$ is

$$\int_{-\infty}^0 dx f(x)\mathrm{Disc}_{x<0}\left(\frac{\log x}{x}\right) = -2\pi i \int_{-\infty}^0 dx \partial_x f(x)\log(-x) \tag{6.15}$$

Using this formula and inputting the boundary CFT data

$$\hat{\gamma}_{\hat{\phi}}^{(1)} = -\alpha \qquad \hat{\mu}^{(1)N} = 0 \qquad a_{\phi^4}^{(1)} = \frac{\alpha}{2} \qquad \gamma_\phi^{(1)} = 0 \qquad \gamma_{\phi^2}^{(1)} = 2\alpha \tag{6.16}$$

we find

$$F^{(1)}(z) = z\left(\alpha + \frac{1}{2 - 2z}\right)\log(1 - z) - \frac{(2\alpha + 1)(z - 2)z\log(z)}{2(z - 1)} \tag{6.17}$$

which agrees with the result of [17]. Here $\alpha$ is a free parameter for the solution of the crossing equation and it can be identified with $\alpha = \frac{1}{2}\frac{N+2}{N+8}$ by matching the result to explicit calculations in the $O(N)$ model. We stress again that the dispersion formula does not provide a simplification in this case because all the (few) boundary CFT data are needed to reconstruct the correlator. The real simplification happens at order $\epsilon^2$.

## 6.2   Order $\epsilon^2$

At second order an infinite number of double trace operators enter the OPE both in the boundary and in the bulk channel. The great advantage of our dispersion relation is that just a finite subset of these operators are sufficient to reconstruct the full correlator. In the bulk channel we have operators with dimensions

$$\Delta_n = \epsilon^2(4 + 2n) + O(\epsilon^3) \tag{6.18}$$

while in the boundary channel we have operators with dimensions

$$\hat{\Delta}_n = \epsilon^2 n + O(\epsilon^3) \tag{6.19}$$

with $n$ an odd integer for Neumann boundary conditions. Naively, we would expect the OPE coefficients of these operators to be necessary to reconstruct the full correlator at this order, however it turns out that the discontinuity does not depend on them and we can reconstruct the correlator without knowing these new operators.

First, we compute the discontinuity at $z = 0$ by looking at the perturbative expansion in the boundary OPE

$$F^{(2)}(z) = (b_{\hat{\Delta}}^2)^{(2)} \hat{f}_{\hat{\Delta}^{(0)}}(z) + ((b_{\hat{\Delta}}^2)^{(0)} \gamma_{\hat{\phi}}^{(2)} + (b_{\hat{\Delta}}^2)^{(1)} \gamma_{\hat{\phi}}^{(1)}) \partial \hat{f}_{\hat{\Delta}^{(0)}}(z)$$
$$+ \frac{1}{2}(b_{\hat{\Delta}}^2)^{(0)} (\gamma_{\hat{\phi}}^{(1)})^2 \partial^2 \hat{f}_{\hat{\Delta}^{(0)}}(z) + \sum_{n=0}^{\infty} (b_{\hat{\Delta}}^2)_n^{(2)} \hat{f}_{\hat{\Delta}_n}(z) \tag{6.20}$$

where we used the short-hand notation $\partial \hat{f}_{\hat{\Delta}^{(0)}}(z) \equiv \partial_{\hat{\Delta}} \hat{f}_{\hat{\Delta}}(z)|_{\hat{\Delta}=\hat{\Delta}^{(0)}}$. The factor $1/z$ in the prefactor (6.7) is always canceled by the blocks or their derivatives, so that the discontinuity is given only by the logarithmic terms arising from the derivatives of the block. These terms can be computed from the OPE data at lower order plus the anomalous dimension of $\hat{\phi}$ at order $\epsilon^2$ which is

$$\gamma_{\hat{\phi}}^{(2)} = \frac{5}{12}\alpha(2\alpha - 1)(4\alpha - 1) \tag{6.21}$$

Notice that the new operators do not contribute. If we evaluate explicitly the perturbative expansion and keep only the terms with logarithms we find

$$\tilde{F}^{(2)}(z) \approx -\frac{5\alpha \left(8\alpha^2 - 6\alpha + 1\right)(z - 2)\log(z)}{12(z-1)^2} - \frac{(2\alpha + 1)^2(z - 2)\log^2(z)}{8(z-1)^2}$$
$$+ \frac{(2\alpha + 1)(2\alpha(z - 1) - 1)\log(1 - z)\log(z)}{4(z-1)^2} + \text{regular at } (z = 0) \tag{6.22}$$

The discontinuity therefore is

$$\text{Disc}_{z<0}(\tilde{F}^{(2)}(z)) = -\frac{5\alpha \left(8\alpha^2 - 6\alpha + 1\right)(z - 2)2\pi i}{12(z-1)^2} - \frac{(2\alpha + 1)^2(z - 2)4\pi i \log(-z)}{8(z-1)^2}$$
$$+ \frac{(2\alpha + 1)(2\alpha(z - 1) - 1)\log(1 - z)2\pi i}{4(z-1)^2} \tag{6.23}$$

The discontinuity at $z = 1$ is more complicated, since the prefactor introduces poles proportional to the identity and $\phi^2$ contributions. However, it is still true that the new operators do not contribute. We can compute the discontinuity using the following finite set of OPE data

$$\gamma_{\phi}^{(2)} = -\frac{1}{12}\alpha(2\alpha - 1) \qquad\qquad \gamma_{\phi^2}^{(2)} = -\frac{1}{12}\alpha(2\alpha - 1)(20\alpha + 3) \tag{6.24}$$

$$\gamma_{\phi^4}^{(2)} = 2 \qquad\qquad a_{\phi^2}^{(2)} = -\frac{1}{3}\alpha(2\alpha - 1)(5\alpha - 1) \tag{6.25}$$

In other words we have reduced the problem of finding infinite coefficients at order $\epsilon^2$ to that of finding only the OPE coefficient of $\phi^2$ and the anomalous dimensions. The latter, moreover, are bulk anomalous dimensions so they can be (and have been) computed in the theory without the boundary. In summary, the only boundary OPE data that we need for

reconstructing the full correlator are $a^{(2)}_{\phi^2}$ and $\gamma^{(2)}_{\hat\phi}$. The computation is analogous to the previous case and yields the following singular terms at $z = 1$

$$
\begin{aligned}
\tilde{F}^{(2)}(z) &\approx \frac{\alpha^2 \mathrm{Li}_2(1-z)}{z-1} - \frac{\pi^2\alpha^2}{6(z-1)} + \frac{\left(1-4\alpha^2(z-1)\right)\log^2(1-z)}{8(z-1)^2} - \frac{\alpha^2(20\alpha(z-1)-21z+23)\log(z)}{12(z-1)^2} \\
&+ \frac{\left(\alpha\left(20\alpha+\pi^2-14\right)+2\right)\alpha}{6(z-1)} - \frac{(4\alpha(z-1)+z-2)\log^2(z)}{8(z-1)^2} + \frac{\alpha(13z-11)\log(z)}{24(z-1)^2} \\
&+ \log(1-z)\left(\frac{\alpha(2\alpha-1)(20\alpha(z-1)-4z+5)}{12(z-1)^2} + \frac{(2\alpha+1)(2\alpha(z-1)-1)\log(z)}{4(z-1)^2}\right) + \text{regular}
\end{aligned}
\tag{6.26}
$$

From this, we can extract the discontinuity at $z = 1$, which will get contributions from the logarithms and the negative powers. By plugging the two discontinuities in the dispersion relation (4.7) and using (6.15) we can compute the correlator. After doing the integral and removing the prefactor we find

$$
\begin{aligned}
F^{(2)}(z) &= \frac{z\log(z)\left(5\alpha\left(8\alpha^2-6\alpha+1\right)(z-2)-3(2\alpha+1)(2\alpha(z-1)-1)\log(1-z)\right)}{12(z-1)} \\
&- \frac{z\log(1-z)\left(\left(3-12\alpha^2(z-1)\right)\log(1-z)+2\alpha(2\alpha-1)(-20\alpha+4(5\alpha-1)z+5)\right)}{24(z-1)} \\
&+ \frac{(2\alpha+1)^2(z-2)z\log^2(z)}{8(z-1)}
\end{aligned}
\tag{6.27}
$$

which corresponds to the result in [20]. One can follow the same procedure for the Dirichlet case and find again perfect agreement with the literature.

## Acknowledgments

We are particularly grateful to V. Forini for collaboration during several stages of this work. We would like to thank J. Barrat, A. Gimenez-Grau and P. Liendo for coordinating the submission of their work with ours. We are also grateful to G. Bliard, E. De Sabbata, M. Lemos, M. Meineri, G. Peveri for useful discussions. The research of LB is funded through the MIUR program for young researchers "Rita Levi Montalcini". The research of DB received partial support through the STFC grant ST/S005803/1.

## A   Kinematics

### A.1   Generic defect $q > 1$

We consider a conformal defect of dimension $p$ and codimension $q$, with $q > 1$, in a $d$-dimensional spacetime. We split the spacetime coordinates into $p$ parallel coordinates $x^a_\parallel$ with $a = q, ..., d-1$ and $q$ orthogonal coordinates $x^i_\perp$ with $i = 0, ... q-1$. The two-point function of bulk identical scalars depends on two conformally invariant cross-ratios and we follow the parametrization of [48], namely

$$
\langle\phi(x_1)\phi(x_2)\rangle = \frac{F(\chi,\eta)}{|x_{1\perp}|^{\Delta_\phi}|x_{2\perp}|^{\Delta_\phi}}
\tag{A.1}
$$

The cross-ratios are related to the lightcone coordinates $(z, \bar{z})$ and the polar coordinates $(r, w)$ used in the main text by

$$\chi = \frac{|x_{12}^{\parallel}|^2 + |x_1^{\perp}|^2 + |x_2^{\perp}|^2}{|x_1^{\perp}||x_2^{\perp}|} = \frac{1 + z\bar{z}}{(z\bar{z})^{\frac{1}{2}}} = \frac{1}{r} + r$$

$$\eta = \frac{x_{1i}x_2^i}{|x_1^{\perp}||x_2^{\perp}|} = \frac{z + \bar{z}}{2(z\bar{z})^{\frac{1}{2}}} = \frac{1}{2}(w + \frac{1}{w})$$

(A.2)

and the lightcone coordinates are related to the radial ones by

$$z = rw$$
$$\bar{z} = \frac{r}{w}$$

(A.3)

The correlator satisfies the crossing equation

$$F(z, \bar{z}) = \sum_{\hat{\Delta},s} b_{\hat{\Delta},s}^2 \, \hat{f}_{\hat{\Delta},s}(z, \bar{z}) = \left( \frac{\sqrt{z\bar{z}}}{(1 - z)(1 - \bar{z})} \right)^{\Delta_\phi} \sum_{\Delta,\ell} a_{\mathcal{O}} \, c_{\phi\phi\mathcal{O}} \, f_{\Delta,\ell}(z, \bar{z}) \,,$$

(A.4)

where the defect conformal blocks are given by

$$\hat{f}_{\hat{\Delta},s}(z, \bar{z}) = z^{\frac{\hat{\Delta}-s}{2}} \bar{z}^{\frac{\hat{\Delta}+s}{2}} {}_2F_1(-s, \frac{q}{2} - 1, 2 - \frac{q}{2} - s, \frac{z}{\bar{z}}) {}_2F_1(\hat{\Delta}, \frac{p}{2}, \hat{\Delta} + 1 - \frac{p}{2}, z\bar{z})$$

(A.5)

They factorize in radial coordinates

$$\hat{f}_{\hat{\Delta},s}(r, w) = \hat{f}_{\hat{\Delta}}(r)\hat{g}_s(w)$$

(A.6)

with

$$\hat{f}_{\hat{\Delta}}(r) = r^{\hat{\Delta}} \, {}_2F_1\left(\hat{\Delta}, \frac{p}{2}, \hat{\Delta} + 1 - \frac{p}{2}, r^2\right) \,, \quad \hat{g}_s(w) = w^{-s} {}_2F_1\left(-s, \frac{q}{2} - 1, 2 - \frac{q}{2} - s, w^2\right) \quad \text{(A.7)}$$

The bulk conformal blocks are not known in a closed form. In [54] they were constructed as a linear combination of two Harish-Chandra functions

$$f_{\Delta,\ell}(z, \bar{z}) = f_{\Delta,\ell}^{HS}(z, \bar{z}) + \frac{\Gamma(\ell + d - 2)\Gamma(-\ell - \frac{d-2}{2})}{\Gamma(\ell + \frac{d-2}{2})\Gamma(-\ell)} \frac{\Gamma(\frac{\ell}{2} + \frac{d-p}{2} - \frac{1}{2})\Gamma(-\frac{\ell}{2} + \frac{1}{2})}{\Gamma(\frac{\ell}{2} + \frac{d}{2} - \frac{1}{2})\Gamma(-\frac{\ell}{2} - \frac{p}{2} + \frac{1}{2})} f_{\Delta,2-d-\ell}^{HS}(z, \bar{z})$$

(A.8)

where $f_{\Delta,\ell}^{HS}(z, \bar{z})$ can be expressed as a double infinite sum

$$f_{\Delta,\ell}^{HS}(z, \bar{z}) = \sum_m^{\infty} \sum_n^{\infty} [(1 - z)(1 - \bar{z})]^{\frac{\Delta-\ell}{2}+m+n} h_n(\Delta, \ell) h_m(1 - \ell, 1 - \Delta) \frac{4^{m-n}}{n!m!} \frac{(\frac{\Delta+\ell}{2})_{n-m}}{(\frac{\Delta+\ell}{2} - \frac{1}{2})_{n-m}}$$
$$\times \, {}_4F_3(-n, -m, \frac{1}{2}, \frac{\Delta-\ell}{2} - \frac{d}{2} + 1; -\frac{\Delta+\ell}{2} + 1 - n, \frac{\Delta+\ell}{2} - m, \frac{\Delta-\ell}{2} - \frac{d}{2} + \frac{3}{2}; 1)$$
$$(1 - z\bar{z})^{\ell-2m} {}_2F_1(\frac{\Delta+\ell}{2} - m + n, \frac{\Delta+\ell}{2} - m + n, \Delta + \ell - 2(m - n), 1 - z\bar{z})$$

(A.9)

where

$$h_n(\Delta, \ell) = \frac{\left(\frac{\Delta}{2} - \frac{1}{2}, \frac{\Delta}{2} - \frac{p}{2}, \frac{\Delta+\ell}{2}\right)_n}{\left(\Delta - \frac{d}{2} + 1, \frac{\Delta+\ell}{2} + \frac{1}{2}\right)_n}$$

(A.10)

## A.2 Boundary q=1

When $q = 1$ the correlator has the form

$$\langle \phi(x_1)\phi(x_2) \rangle = \frac{F(\xi)}{(4|x_2^\perp||x_2^\perp|)^{\Delta_\phi}} \tag{A.11}$$

and depends only on the cross-ratio

$$\xi = \frac{(x_1 - x_2)^2}{4x_1^\perp x_2^\perp} \tag{A.12}$$

where $\xi > 0$ when the two operators live in the Euclidean signature or are spacelike separated in the Lorentzian signature. The boundary block expansion, corresponding to sending $\xi \to \infty$ reads

$$F(\xi) = \sum_{\hat{\Delta}} \hat{b}_{\hat{\Delta}}^2 \hat{f}_{\hat{\Delta}}(\xi) \tag{A.13}$$

where the sum runs over the dimensions of the defect operators $\hat{\Delta}$, $\hat{b}_{\hat{\Delta}}$ are real OPE coefficients and the conformal block is

$$\hat{f}_{\hat{\Delta}}(\xi) = (\xi)^{-\hat{\Delta}} {}_2F_1\left( \hat{\Delta}, \hat{\Delta} + 1 - \frac{d}{2}, 2\hat{\Delta} + 2 - d, -\frac{1}{\xi} \right) \tag{A.14}$$

The bulk expansion, obtained by sending $\xi \to 0$ is

$$F(\xi) = (\xi)^{-\Delta_\phi} \sum_{\Delta} a_{\mathcal{O}} c_{\phi\phi\mathcal{O}} f_\Delta(\xi) \tag{A.15}$$

where the sum runs over the dimensions of the bulk operators $\Delta$ and

$$f_\Delta(\xi) = (\xi)^{\frac{\Delta}{2}} {}_2F_1\left( \frac{\Delta}{2}, \frac{\Delta}{2}, \Delta + 1 - \frac{d}{2}, -\xi \right) \tag{A.16}$$

Here $a_{\mathcal{O}}$ are the one-point function coefficients (which are non zero in presence of a defect) and $c_{\phi\phi\mathcal{O}}$ are the OPE coefficients. The relation between the variable $\xi$ and the variable $z$ used in the main text is

$$z = \frac{1}{\xi + 1} . \tag{A.17}$$

## B  Superblocks

Here we review the superconformal blocks that were found in [43] and that we used in Section 5. After defining the R-symmetry block

$$h_k = \sigma^{-\frac{k}{2}} {}_2F_1\left( -\frac{k}{2}, -\frac{k}{2}; -k - 1; \frac{\sigma}{2} \right) \tag{B.1}$$

then the superconformal blocks are expressed as a combination of ordinary bulk blocks $f_{\Delta,\ell}$ (4.5) as

$$\mathcal{G}_{\mathcal{B}_{[0,P,0]}} = h_P f_{P,0}(z,\bar{z}) + \frac{(P+2)^2 P}{128(P+1)^2(P+3)} h_{P-2} f_{P+2,2}(z,\bar{z})$$

$$+ \frac{(P-2)(P+2)P^2}{16384(P-1)^2(P+1)(P+3)} h_{P-4} f_{P+4,0}(z,\bar{z})$$

$$\mathcal{G}_{\mathcal{C}_{[0,2,0],\ell}} = h_2 f_{\ell+4,\ell} + b_1 h_0 f_{\ell+6,\ell-2} + (b_{21} h_4 + b_{22} h_2 + b_{23} h_0) f_{\ell+6,\ell+2} + (b_{31} h_2 + b_{32} h_0) f_{\ell+8,\ell}$$

$$+ b_4 h_2 f_{\ell+8,\ell+4} + b_5 h_0 f_{\ell+10,\ell+2}$$

$$\mathcal{G}_{\mathcal{A}_{[0,0,0],\ell}} = h_0 f_{\Delta,\ell} + (h_2 \eta_{11} + h_0 \eta_{12}) f_{\Delta+2,\ell-2} + (h_2 \eta_{21} + h_0 \eta_{22}) f_{\Delta+2,\ell+2} + \eta_3 h_0 f_{\Delta+4,\ell-4}$$

$$+ (h_4 \eta_{41} + h_2 \eta_{42} + h_0 \eta_{43}) f_{\Delta+4,\ell} + \eta_5 h_0 f_{\Delta+4,\ell+4} + (h_2 \eta_{61} + h_0 \eta_{62}) f_{\Delta+6,\ell-2} +$$

$$(h_2 \eta_{71} + h_0 \eta_{72}) f_{\Delta+6,\ell+2} + \eta_8 h_0 f_{\Delta+8,\ell}$$

$$\text{(B.2)}$$

where the explicit coefficients can be found in [43].

## C Dispersion relation from Lorentzian inversion formula

Instead of using Cauchy's theorem, we can obtain a dispersion relation starting from the conformal partial wave expansion (2.9) and inserting the coefficient function $b(\hat{,}s)$ obtained through the Lorentzian inversion formula (2.12). Unsurprisingly, after exchanging the order of integration, we find an expression of the form

$$F(r,w) = \int_0^1 d\tilde{r} \int_0^{\tilde{r}} d\tilde{w} \; K(\tilde{r},\tilde{w},r,w) \mathrm{Disc} F(\tilde{r},\tilde{w}) \qquad \text{(C.1)}$$

with

$$K(\tilde{r},\tilde{w},r,w) = S(w,\tilde{w}) I(r,\tilde{r})$$

$$S(w,\tilde{w}) = \sum_{s=0}^{\infty} \tilde{w}^{1-q}(1-\tilde{w}^2)^{q-2} \hat{g}_{2-q-s}(\tilde{w}) \hat{g}_s(w) \qquad \text{(C.2)}$$

$$I(r,\tilde{r}) = \int_{\frac{p}{2}-i\infty}^{\frac{p}{2}+i\infty} \frac{d\hat{\Delta}}{2\pi i} \; \tilde{r}^{-p-1}(1-\tilde{r}^2)^p \frac{\hat{K}_{\hat{\Delta}}}{i\pi \hat{K}_{p-\hat{\Delta}}} \; \hat{\Psi}_{\hat{\Delta}}(\tilde{r}) \Psi_{\hat{\Delta}}(r)$$

Notice that the contributions from the angular and radial part factorize. The angular contribution $S(w,\tilde{w})$ can be computed using integral representations of the Gegenbauer polynomials and of the hypergeometric function and exchanging the sum over spin with the integrals coming from the integral representations. The result is

$$S(w,\tilde{w}) = \frac{1}{2\pi i} \left( \frac{1}{\tilde{w}-w} + \frac{1}{\tilde{w}-\frac{1}{w}} - \frac{1}{\tilde{w}} \right) \qquad \text{(C.3)}$$

The remaining contribution $I(r,\tilde{r})$ can be evaluated exploiting the orthogonality of conformal partial waves, namely

$$\int_0^1 dr \; r^{-p-1}(1-r^2)^p \Psi_{\hat{\Delta}_1}(r) \Psi_{\hat{\Delta}_2}(r) = \frac{\pi}{2} \frac{K_{p-\hat{\Delta}_2}}{K_{\hat{\Delta}_1}} (\delta(\nu_1-\nu_2) + \delta(\nu_1+\nu_2)) \qquad \text{(C.4)}$$

where $\Delta = \frac{1}{2} + i\nu$, and the shadow conformal partial wave

$$\Psi_{p-\hat{\Delta}}(r) = \frac{K_{\hat{\Delta}}}{K_{p-\hat{\Delta}}}\Psi_{\hat{\Delta}}(r) \tag{C.5}$$

If we introduce a generic function $f(r)$ and expand it on the $\Psi_{\hat{\Delta}}(r)$ basis

$$f(r) = \int_{\frac{p}{2}-i\infty}^{\frac{p}{2}+i\infty} \frac{d\Delta'}{2\pi i} \hat{f}(\Delta')\Psi_{\Delta'}(r) \tag{C.6}$$

then, using the properties above, we find

$$\int_0^1 dr\, I(r, \tilde{r}) f(\tilde{r})$$
$$= \int_0^1 dr\, I(r, \tilde{r}) \int_{\frac{1}{2}-i\infty}^{\frac{1}{2}+i\infty} \frac{d\Delta'}{2\pi i} \hat{f}(\Delta')\Psi_{\Delta'}(\tilde{r})$$
$$= \int_{\frac{p}{2}-i\infty}^{\frac{p}{2}+i\infty} \frac{d\hat{\Delta}}{2\pi i} \int_{\frac{p}{2}-i\infty}^{\frac{p}{2}+i\infty} \frac{d\Delta'}{2\pi i} \frac{K_{\hat{\Delta}}}{K_{p-\hat{\Delta}}} \Psi_{\hat{\Delta}}(r)\hat{f}(\Delta') \int_0^1 d\tilde{r}\, \tilde{r}^{-p-1}(1-\tilde{r}^2)^p \Psi_{\hat{\Delta}}(\tilde{r})\Psi_{\Delta'}(\tilde{r}) \tag{C.7}$$
$$= \int_{\frac{p}{2}-i\infty}^{\frac{p}{2}+i\infty} \frac{d\hat{\Delta}}{2\pi i} \int_{\frac{p}{2}-i\infty}^{\frac{p}{2}+i\infty} \frac{d\Delta'}{2\pi i} \frac{K_{\hat{\Delta}}}{K_{p-\hat{\Delta}}} \Psi_{\hat{\Delta}}(r)\hat{f}(\Delta')\frac{\pi}{2}\frac{K_{p-\Delta'}}{K_{\hat{\Delta}}}(\delta(\hat{\nu}-\nu') + \delta(\hat{\nu}+\nu'))$$
$$= \int_{\frac{p}{2}-i\infty}^{\frac{p}{2}+i\infty} \frac{d\hat{\Delta}}{2\pi i}\Psi_{\hat{\Delta}}(r)\hat{f}(\hat{\Delta})$$
$$= f(r)$$

which implies

$$I(r, \tilde{r}) = \delta(r - \tilde{r}) \tag{C.8}$$

We can finally collect the pieces (C.3), (C.8) and we find our dispersion relation

$$F(r, w) = \int_0^1 d\tilde{r} \int_0^{\tilde{r}} d\tilde{w}\, \text{Disc} F(\tilde{r}, \tilde{w})\frac{1}{2\pi i}\left(\frac{1}{\tilde{w}-w} + \frac{1}{\tilde{w}-\frac{1}{w}} - \frac{1}{\tilde{w}}\right)\delta(r - \tilde{r}) \tag{C.9}$$

which is exactly the formula we found using Cauchy's theorem.

# D Results for $F_{5,p}^{(1)}$ with $p \leq 3$

$$F_{5,0}^{(1)} = -\frac{5}{4}\left(\frac{r^4\sqrt{r^2}\left(r^2+1\right)w^4}{\left(r^2-1\right)^2(r-w)^4(rw-1)^4} - \frac{2r^6\sqrt{r^2}w^4\log\left(r^2\right)}{\left(r^2-1\right)^3(r-w)^4(rw-1)^4}\right)$$

$$F_{5,1}^{(1)} = \frac{5}{4}\left(\frac{\left(r^4-38r^2+1\right)r^5w^3}{\left(r^2-1\right)^4(r-w)^3(rw-1)^3}\right.$$
$$\left. + \frac{r^6w^3\left(r^6\left(w^2+1\right)+5r^5w-10r^4\left(w^2+1\right)+26r^3w-10r^2\left(w^2+1\right)+5rw+w^2+1\right)\log\left(r^2\right)}{\left(r^2-1\right)^5(r-w)^4(rw-1)^4}\right)$$

$$F_{5,2}^{(1)} = \frac{5}{4}\left(-\frac{3r^6w^2\left(8r^6\left(w^2+1\right)+43r^5w-83r^4\left(w^2+1\right)+214r^3w-83r^2\left(w^2+1\right)+43rw+8w^2+8\right)}{\left(r^2-1\right)^6(r-w)^3(rw-1)^3}\right.$$
$$+ \frac{3r^5w^2\left(r\left(r^9w+7r^8\left(w^2+1\right)+15r^7w-46r^6\left(w^2+1\right)+284r^5w\right)+w\right)\log\left(r^2\right)}{2\left(r^2-1\right)^7(r-w)^3(rw-1)^3}$$
$$\left. + \frac{3r^5w^2\left(r\left(-222r^4\left(w^2+1\right)+284r^3w-46r^2\left(w^2+1\right)+15rw+7w^2+7\right)+w\right)\log\left(r^2\right)}{2\left(r^2-1\right)^7(r-w)^3(rw-1)^3}\right)$$

$$F_{5,3}^{(1)} = \frac{5}{4}\left(\frac{5r^5w\left(r\left(3r^{11}w+15r^{10}\left(w^2+1\right)+20r^9w-35r^8\left(w^2+1\right)+833r^7w-960r^6\left(w^2+1\right)\right)+3w\right)\log\left(r^2\right)}{2\left(r^2-1\right)^9(r-w)^2(rw-1)^2}\right.$$
$$+ \frac{5r^5w\left(r\left(+2208r^5w-960r^4\left(w^2+1\right)+833r^3w-35r^2\left(w^2+1\right)+20rw+15\left(w^2+1\right)\right)+3w\right)\log\left(r^2\right)}{2\left(r^2-1\right)^9(r-w)^2(rw-1)^2}$$
$$- \frac{5r^5w\left(r\left(9r^9w+72r^8\left(w^2+1\right)+188r^7w-503r^6\left(w^2+1\right)+2743r^5w-2078r^4\left(w^2+1\right)+2743r^3w\right)+9w\right)}{3\left(r^2-1\right)^8(r-w)^2(rw-1)^2}$$
$$\left. - \frac{5r^5w\left(r\left(-503r^2\left(w^2+1\right)+188rw+72\left(w^2+1\right)\right)+9w\right)}{3\left(r^2-1\right)^8(r-w)^2(rw-1)^2}\right)$$

(D.1)

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
