# Peer review of "Conformal dispersion relations for defects and boundaries"

_SciPost Physics_

## Round 1 · Referee Report · Aleix Gimenez-Grau · 2022-9-29

Strengths
1 - The paper introduces a new dispersion relation for defect CFT, which is a simple and powerful tool to bootstrap defect CFTs analytically.
2 - The dispersion relation is applied to several examples, which show that with the new technology one can easily rederive complicated results from the literature.
Weaknesses
1 - In the BCFT example, the authors do not use the full power of crossing symmetry. They then need to input more CFT data than necessary to fix their result, obscuring the power of the dispersion relation.
2 - The example of N=4 SYM at weak coupling requires rewriting to be clearer.
3 - Some sections contain minor errors, mostly sign mistakes. This can make it hard for readers to reproduce the calculations.
Report
Analytic studies of defect CFT have recently attracted a lot of interest, and this paper introduces a powerful tool that considerably simplifies the calculations in this field. As a result, I believe this technique will be used extensively in the community in future work. Furthermore, the paper contains a detailed introduction and is written in a clear way.
However, I do believe the discussion in sections 5.3 and 6 should be improved to show the effectiveness of the dispersion relation, and the paper can be accepted for publication provided these issues are addressed.
I would also like to apologize to the authors and editor for the late report.
Requested changes
Major changes:
1 - As it is written now, section 5.3 seems incorrect to me, but perhaps after a rewriting these issues will be clarified:
1.1 - I don't understand the argument that only short multiplets contribute at leading order. There are longs below twist-four, see [41, (4.17b)], and they should contribute to Disc.
1.2 - Even if only B[0,2,0] contributes, I don't understand equation (5.25). The discontinuity of B[0,2,0] was computed in [42] and is different than (5.25). Furthermore, B[0,2,0] should give a contribution to the $F_0$ channel.
1.3 - At next-to-leading order, the authors claim "We can write down these terms as infinite sums but we cannot evaluate them because the bulk blocks are not known in closed form". I think it is a very complicated problem to write down this infinite sum "a priori". Indeed, one would need knowledge of bulk anomalous dimensions and one would need to solve a complicated mixing problem.
1.4 - Also at next-to-leading order, the authors "compute" $F^{(2)}_{2,0}$. However, my understanding is that they are actually expanding the result of [42] in blocks, and then they apply the dispersion relation term by term. However, this is not a first-principle derivation of $F^{(2)}_{2,0}$.
1.5 - Summarizing, I believe the authors are not rederiving [42], but instead using the results in [42] as input to their dispersion relation. I suggest that the section is rewritten to make clear that this is the logic.
2 - Although section 6 is correct, I believe it can be improved. My main objection is that the authors make unnecessary assumptions that make the bootstrap calculation look less powerful:
2.1 - In eq. (6.9), you don't need to assume that only $\hat\phi$ contributes to $F^{(1)}$. In fact there could be infinitely many defect operators at order 1, see for example 2012.00018. The point is that even if they were there the discontinuity would kill them. A posteriori, when you have bootstrapped the full correlator, you will have a "proof" that the operators were not there to start with.
2.2 - In eq. (6.14), again the most general ansatz for $F^{(1)}$ should contain infinitely many bulk operators, and it is non-trivial that in the end they don't contribute. If you don't include them, you should justify it from (3.1) in 1712.02314.
2.3 - In eq. (6.17), there is no need to input any CFT data. You can reconstruct the correlator from the dispersion relation, and get a formula that depends on $\hat\gamma_{\hat\phi}$, $a^{(1)}_{\phi^2}$ and $\gamma_{\phi^2}$. Demanding that this has a consistent bulk and defect expansion fixes $\hat\gamma_{\hat\phi}$ and $a^{(1)}_{\phi^2}$. The only necessary input is $\gamma_{\phi^2}$, which is known from the theory without defect.
2.4 - Although I have not done the calculation, similar comments should apply to order $\epsilon^2$. I believe you only need the dispersion relation, crossing and boundary-independent information like $\gamma_\phi$, $\gamma_{\phi^2}$ and $\gamma_{\phi^4}$, to fix the full $O(\epsilon^2)$ result.
Minor changes:
1 - p.7: "close form" -> "closed form"
2 - p. 10: "If this is case" -> "If this is the case"
3 - p. 13: "To strategy" -> "The strategy"
4 - p. 13: Provide a reference for equation (3.16), probably [52].
5 - p. 13: "computed it" -> "computed"
6 - p. 13: w is never introduced.
7 - p. 13: Explain why they include the factors u/v in eq. (3.18).
8 - p. 15: The minus sign in eq. (4.7) should be a + sign. This error is also in (4.10).
9 - p. 16: The discontinuity (4.12) has the incorrect sign.
10 - p. 16: Eq. (4.13) should have $\delta(z)$ -> $\delta(z')$. The signs need to be fixed.
11 - p. 16: Eq. (4.15) should read $(-z)^{\Delta_\phi-1}$, the sign in the second line is wrong.
12 - p. 17: Eq. (4.16) many primes are missing. Also the powers should read $(-z')^{\Delta_\phi-1}$. The signs need to be fixed as discussed above.
13 - p. 18: Seection (3) -> Section 3
14 - p. 19: "At order 1/N^2 we have two contributions". This is somewhat misleading, since there are infinitely $1/\sqrt{\lambda}$ corrections.
15 - p. 19: "dispersion relation (C.9)" probably should read "dispersion relation (3.3)".
16 - p. 21: Eq. (5.24) is incorrect, it should have r,w dependance.
17 - p. 22: "superblock boundary expansion" should probably read "superblock bulk expansion"
18 - p. 22: "With [41]" -> "With [41]."
19 - p. 22: "Form" -> "From"
20 - p. 23: $c_{\phi\phi O}$ should read $\lambda_{\phi\phi O}$ for consistency
20 - p. 23: It is unfortunate to redefine $a = a_O \lambda$, since it makes equations such as (6.11) ambiguous.
21 - p. 28: 0, ..., q-1 lacks a space
22 - p. 30: Before [41], these blocks were derived in [29]. Also, the authors seem to be using the conventions from [42], so they should probably mention it.
23 - p. 31: $b(\hat,s)$ should be fixed
24 - It would be good if the authors review minus signs related to the BCFT dispersion relation throughout the paper.
Author: Lorenzo Bianchi on 2023-01-16 [id 3239]
(in reply to Report 1 by Aleix Gimenez-Grau on 2022-09-29)We would like to thank the referee for the thorough report and for the comments. We modified sections 5.3 and 6 according to the requests and implemented the minor corrections. A more comprehensive list of changes will appear with the resubmission.

---

## Editorial Decision

submission_&_refereeing_history